# Less is More: Nyström Computational Regularization

**Alessandro Rudi**[†]     **Raffaello Camoriano**[†‡]     **Lorenzo Rosasco**[†∘]

[†]Università degli Studi di Genova - DIBRIS, Via Dodecaneso 35, Genova, Italy

[‡]Istituto Italiano di Tecnologia - iCub Facility, Via Morego 30, Genova, Italy

[∘]Massachusetts Institute of Technology and Istituto Italiano di Tecnologia
Laboratory for Computational and Statistical Learning, Cambridge, MA 02139, USA

{ale_rudi, lrosasco}@mit.edu     raffaello.camoriano@iit.it

## Abstract

We study Nyström type subsampling approaches to large scale kernel methods, and prove learning bounds in the statistical learning setting, where random sampling and high probability estimates are considered. In particular, we prove that these approaches can achieve optimal learning bounds, provided the subsampling level is suitably chosen. These results suggest a simple incremental variant of Nyström Kernel Regularized Least Squares, where the subsampling level implements a form of computational regularization, in the sense that it controls at the same time regularization and computations. Extensive experimental analysis shows that the considered approach achieves state of the art performances on benchmark large scale datasets.

## 1 Introduction

Kernel methods provide an elegant and effective framework to develop nonparametric statistical approaches to learning [1]. However, memory requirements make these methods unfeasible when dealing with large datasets. Indeed, this observation has motivated a variety of computational strategies to develop large scale kernel methods [2–8].

In this paper we study subsampling methods, that we broadly refer to as Nyström approaches. These methods replace the empirical kernel matrix, needed by standard kernel methods, with a smaller matrix obtained by (column) subsampling [2, 3]. Such procedures are shown to often dramatically reduce memory/time requirements while preserving good practical performances [9–12]. The goal of our study is two-fold. First, and foremost, we aim at providing a theoretical characterization of the generalization properties of such learning schemes in a statistical learning setting. Second, we wish to understand the role played by the subsampling level both from a statistical and a computational point of view. As discussed in the following, this latter question leads to a natural variant of Kernel Regularized Least Squares (KRLS), where the subsampling level controls both regularization and computations.

From a theoretical perspective, the effect of Nyström approaches has been primarily characterized considering the discrepancy between a given empirical kernel matrix and its subsampled version [13–19]. While interesting in their own right, these latter results do not directly yield information on the generalization properties of the obtained algorithm. Results in this direction, albeit suboptimal, were first derived in [20] (see also [21,22]), and more recently in [23,24]. In these latter papers, sharp error analyses in expectation are derived in a fixed design regression setting for a form of Kernel Regularized Least Squares. In particular, in [23] a basic uniform sampling approach is studied, while in [24] a subsampling scheme based on the notion of leverage score is considered. The main technical contribution of our study is an extension of these latter results to the statistical learning setting, where the design is random and high probability estimates are considered. The

more general setting makes the analysis considerably more complex. Our main result gives optimal finite sample bounds for both uniform and leverage score based subsampling strategies. These methods are shown to achieve the same (optimal) learning error as kernel regularized least squares, recovered as a special case, while allowing substantial computational gains. Our analysis highlights the interplay between the regularization and subsampling parameters, suggesting that the latter can be used to control simultaneously regularization and computations. This strategy implements a form of *computational regularization* in the sense that the computational resources are tailored to the generalization properties in the data. This idea is developed considering an incremental strategy to efficiently compute learning solutions for different subsampling levels. The procedure thus obtained, which is a simple variant of classical Nyström Kernel Regularized Least Squares with uniform sampling, allows for efficient model selection and achieves state of the art results on a variety of benchmark large scale datasets.

The rest of the paper is organized as follows. In Section 2, we introduce the setting and algorithms we consider. In Section 3, we present our main theoretical contributions. In Section 4, we discuss computational aspects and experimental results.

## 2   Supervised learning with KRLS and Nyström approaches

Let $X \times \mathbb{R}$ be a probability space with distribution $\rho$, where we view $X$ and $\mathbb{R}$ as the input and output spaces, respectively. Let $\rho_X$ denote the marginal distribution of $\rho$ on $X$ and $\rho(\cdot|x)$ the conditional distribution on $\mathbb{R}$ given $x \in X$. Given a hypothesis space $\mathcal{H}$ of measurable functions from $X$ to $\mathbb{R}$, the goal is to minimize the *expected risk*,

$$\min_{f \in \mathcal{H}} \mathcal{E}(f), \qquad \mathcal{E}(f) = \int_{X \times \mathbb{R}} (f(x) - y)^2 d\rho(x, y), \tag{1}$$

provided $\rho$ is known only through a training set of $(x_i, y_i)_{i=1}^n$ sampled identically and independently according to $\rho$. A basic example of the above setting is random design regression with the squared loss, in which case

$$y_i = f_*(x_i) + \epsilon_i, \quad i = 1, \ldots, n, \tag{2}$$

with $f_*$ a fixed *regression* function, $\epsilon_1, \ldots, \epsilon_n$ a sequence of random variables seen as noise, and $x_1, \ldots, x_n$ random inputs. In the following, we consider kernel methods, based on choosing a hypothesis space which is a separable reproducing kernel Hilbert space. The latter is a Hilbert space $\mathcal{H}$ of functions, with inner product $\langle \cdot, \cdot \rangle_{\mathcal{H}}$, such that there exists a function $K : X \times X \to \mathbb{R}$ with the following two properties: 1) for all $x \in X$, $K_x(\cdot) = K(x, \cdot)$ belongs to $\mathcal{H}$, and 2) the so called reproducing property holds: $f(x) = \langle f, K_x \rangle_{\mathcal{H}}$, for all $f \in \mathcal{H}$, $x \in X$ [25]. The function $K$, called reproducing kernel, is easily shown to be symmetric and positive definite, that is the kernel matrix $(K_N)_{i,j} = K(x_i, x_j)$ is positive semidefinite for all $x_1, \ldots, x_N \in X$, $N \in \mathbb{N}$. A classical way to derive an empirical solution to problem (1) is to consider a Tikhonov regularization approach, based on the minimization of the penalized empirical functional,

$$\min_{f \in \mathcal{H}} \frac{1}{n} \sum_{i=1}^n (f(x_i) - y_i)^2 + \lambda \|f\|_{\mathcal{H}}^2, \lambda > 0. \tag{3}$$

The above approach is referred to as Kernel Regularized Least Squares (KRLS) or Kernel Ridge Regression (KRR). It is easy to see that a solution $\hat{f}_\lambda$ to problem (3) exists, it is unique and the representer theorem [1] shows that it can be written as

$$\hat{f}_\lambda(x) = \sum_{i=1}^n \hat{\alpha}_i K(x_i, x) \quad \text{with} \quad \hat{\alpha} = (K_n + \lambda n I)^{-1} y, \tag{4}$$

where $x_1, \ldots, x_n$ are the training set points, $y = (y_1, \ldots, y_n)$ and $K_n$ is the empirical kernel matrix. Note that this result implies that we can restrict the minimization in (3) to the space,

$$\mathcal{H}_n = \{f \in \mathcal{H} \mid f = \sum_{i=1}^n \alpha_i K(x_i, \cdot), \ \alpha_1, \ldots, \alpha_n \in \mathbb{R}\}.$$

Storing the kernel matrix $K_n$, and solving the linear system in (4), can become computationally unfeasible as $n$ increases. In the following, we consider strategies to find more efficient solutions,

based on the idea of replacing $\mathcal{H}_n$ with

$$\mathcal{H}_m = \{f \mid f = \sum_{i=1}^{m} \alpha_i K(\tilde{x}_i, \cdot), \ \alpha \in \mathbb{R}^m\},$$

where $m \leq n$ and $\{\tilde{x}_1, \ldots, \tilde{x}_m\}$ is a subset of the input points in the training set. The solution $\hat{f}_{\lambda,m}$ of the corresponding minimization problem can now be written as,

$$\hat{f}_{\lambda,m}(x) = \sum_{i=1}^{m} \tilde{\alpha}_i K(\tilde{x}_i, x) \quad \text{with} \quad \tilde{\alpha} = (K_{nm}^\top K_{nm} + \lambda n K_{mm})^\dagger K_{nm}^\top y, \tag{5}$$

where $A^\dagger$ denotes the Moore-Penrose pseudoinverse of a matrix $A$, and $(K_{nm})_{ij} = K(x_i, \tilde{x}_j)$, $(K_{mm})_{kj} = K(\tilde{x}_k, \tilde{x}_j)$ with $i \in \{1, \ldots, n\}$ and $j, k \in \{1, \ldots, m\}$ [2]. The above approach is related to Nyström methods and different approximation strategies correspond to different ways to select the inputs subset. While our framework applies to a broader class of strategies, see Section C.1, in the following we primarily consider two techniques.
**Plain Nyström**. The points $\{\tilde{x}_1, \ldots, \tilde{x}_m\}$ are sampled uniformly at random without replacement from the training set.
**Approximate leverage scores (ALS) Nyström**. Recall that the *leverage scores* associated to the training set points $x_1, \ldots, x_n$ are

$$(l_i(t))_{i=1}^n, \quad l_i(t) = (K_n(K_n + tnI)^{-1})_{ii}, \quad i \in \{1, \ldots, n\} \tag{6}$$

for any $t > 0$, where $(K_n)_{ij} = K(x_i, x_j)$. In practice, leverage scores are onerous to compute and approximations $(\hat{l}_i(t))_{i=1}^n$ can be considered [16, 17, 24]. In particular, in the following we are interested in suitable approximations defined as follows:

**Definition 1** ($T$-approximate leverage scores)**.** *Let $(l_i(t))_{i=1}^n$ be the leverage scores associated to the training set for a given $t$. Let $\delta > 0$, $t_0 > 0$ and $T \geq 1$. We say that $(\hat{l}_i(t))_{i=1}^n$ are $T$-approximate leverage scores with confidence $\delta$, when with probability at least $1 - \delta$,*

$$\frac{1}{T} l_i(t) \leq \hat{l}_i(t) \leq T l_i(t) \quad \forall i \in \{1, \ldots, n\}, t \geq t_0.$$

Given $T$-approximate leverage scores for $t > \lambda_0$, $\{\tilde{x}_1, \ldots, \tilde{x}_m\}$ are sampled from the training set independently with replacement, and with probability to be selected given by $P_t(i) = \hat{l}_i(t)/\sum_j \hat{l}_j(t)$. In the next section, we state and discuss our main result showing that the KRLS formulation based on plain or approximate leverage scores Nyström provides optimal empirical solutions to problem (1).

## 3 Theoretical analysis

In this section, we state and discuss our main results. We need several assumptions. The first basic assumption is that problem (1) admits at least a solution.

**Assumption 1.** *There exists an $f_{\mathcal{H}} \in \mathcal{H}$ such that*

$$\mathcal{E}(f_{\mathcal{H}}) = \min_{f \in \mathcal{H}} \mathcal{E}(f).$$

Note that, while the minimizer might not be unique, our results apply to the case in which $f_{\mathcal{H}}$ is the unique minimizer with minimal norm. Also, note that the above condition is weaker than assuming the regression function in (2) to belong to $\mathcal{H}$. Finally, we note that the study of the paper can be adapted to the case in which minimizers do not exist, but the analysis is considerably more involved and left to a longer version of the paper.
The second assumption is a basic condition on the probability distribution.

**Assumption 2.** *Let $z_x$ be the random variable $z_x = y - f_{\mathcal{H}}(x)$, with $x \in X$, and $y$ distributed according to $\rho(y|x)$. Then, there exists $M, \sigma > 0$ such that $\mathbb{E}|z_x|^p \leq \frac{1}{2}p!M^{p-2}\sigma^2$ for any $p \geq 2$, almost everywhere on $X$.*

The above assumption is needed to control random quantities and is related to a *noise* assumption in the regression model (2). It is clearly weaker than the often considered bounded output assumption

[25], and trivially verified in classification.

The last two assumptions describe the capacity (roughly speaking the *"size"*) of the hypothesis space induced by $K$ with respect to $\rho$ and the regularity of $f_{\mathcal{H}}$ with respect to $K$ and $\rho$. To discuss them, we first need the following definition.

**Definition 2** (Covariance operator and effective dimensions). *We define the covariance operator as*

$$C : \mathcal{H} \to \mathcal{H}, \quad \langle f, Cg \rangle_{\mathcal{H}} = \int_X f(x)g(x)d\rho_X(x) \ , \quad \forall f, g \in \mathcal{H}.$$

*Moreover, for $\lambda > 0$, we define the random variable $\mathcal{N}_x(\lambda) = \langle K_x, (C + \lambda I)^{-1} K_x \rangle_{\mathcal{H}}$ with $x \in X$ distributed according to $\rho_X$ and let*

$$\mathcal{N}(\lambda) = \mathbb{E} \, \mathcal{N}_x(\lambda), \qquad \mathcal{N}_\infty(\lambda) = \sup_{x \in X} \mathcal{N}_x(\lambda).$$

We add several comments. Note that $C$ corresponds to the second moment operator, but we refer to it as the covariance operator with an abuse of terminology. Moreover, note that $\mathcal{N}(\lambda) = \operatorname{Tr}(C(C + \lambda I)^{-1})$ (see [26]). This latter quantity, called effective dimension or degrees of freedom, can be seen as a measure of the capacity of the hypothesis space. The quantity $\mathcal{N}_\infty(\lambda)$ can be seen to provide a uniform bound on the leverage scores in Eq. (6). Clearly, $\mathcal{N}(\lambda) \leq \mathcal{N}_\infty(\lambda)$ for all $\lambda > 0$.

**Assumption 3.** *The kernel $K$ is measurable, $C$ is bounded. Moreover, for all $\lambda > 0$ and a $Q > 0$,*

$$\mathcal{N}_\infty(\lambda) < \infty, \tag{7}$$

$$\mathcal{N}(\lambda) \leq Q\lambda^{-\gamma}, \quad 0 < \gamma \leq 1. \tag{8}$$

Measurability of $K$ and boundedness of $C$ are minimal conditions to ensure that the covariance operator is a well defined linear, continuous, self-adjoint, positive operator [25]. Condition (7) is satisfied if the kernel is bounded $\sup_{x \in X} K(x, x) = \kappa^2 < \infty$, indeed in this case $\mathcal{N}_\infty(\lambda) \leq \kappa^2/\lambda$ for all $\lambda > 0$. Conversely, it can be seen that condition (7) together with boundedness of $C$ imply that the kernel is bounded, indeed [1]

$$\kappa^2 \leq 2\|C\|\mathcal{N}_\infty(\|C\|).$$

Boundedness of the kernel implies in particular that the operator $C$ is trace class and allows to use tools from spectral theory. Condition (8) quantifies the capacity assumption and is related to covering/entropy number conditions (see [25] for further details). In particular, it is known that condition (8) is ensured if the eigenvalues $(\sigma_i)_i$ of $C$ satisfy a polynomial decaying condition $\sigma_i \sim i^{-\frac{1}{\gamma}}$. Note that, since the operator $C$ is trace class, Condition (8) always holds for $\gamma = 1$. Here, for space constraints and in the interest of clarity we restrict to such a polynomial condition, but the analysis directly applies to other conditions including exponential decay or a finite rank conditions [26]. Finally, we have the following regularity assumption.

**Assumption 4.** *There exists $s \geq 0$, $1 \leq R < \infty$, such that $\|C^{-s} f_{\mathcal{H}}\|_{\mathcal{H}} < R$.*

The above condition is fairly standard, and can be equivalently formulated in terms of classical concepts in approximation theory such as interpolation spaces [25]. Intuitively, it quantifies the degree to which $f_{\mathcal{H}}$ can be well approximated by functions in the RKHS $\mathcal{H}$ and allows to control the bias/approximation error of a learning solution. For $s = 0$, it is always satisfied. For larger $s$, we are assuming $f_{\mathcal{H}}$ to belong to subspaces of $\mathcal{H}$ that are the images of the fractional compact operators $C^s$. Such spaces contain functions which, expanded on a basis of eigenfunctions of $C$, have larger coefficients in correspondence to large eigenvalues. Such an assumption is natural in view of using techniques such as (4), which can be seen as a form of spectral filtering, that estimate stable solutions by discarding the contribution of small eigenvalues [27]. In the next section, we are going to quantify the quality of empirical solutions of Problem (1) obtained by schemes of the form (5), in terms of the quantities in Assumptions 2, 3, 4.

### 3.1 Main results

In this section, we state and discuss our main results, starting with optimal finite sample error bounds for regularized least squares based on plain and approximate leverage score based Nyström subsampling.

**Theorem 1.** *Under Assumptions 1, 2, 3, and 4, let $\delta > 0$, $v = \min(s, 1/2)$, $p = 1 + 1/(2v + \gamma)$ and assume*

$$n \geq 1655\kappa^2 + 223\kappa^2 \log \frac{6\kappa^2}{\delta} + \left( \frac{38p}{\|C\|} \log \frac{114\kappa^2 p}{\|C\|\delta} \right)^p.$$

*Then, the following inequality holds with probability at least $1 - \delta$,*

$$\mathcal{E}(\hat{f}_{\lambda,m}) - \mathcal{E}(f_{\mathcal{H}}) \leq q^2 \, n^{-\frac{2v+1}{2v+\gamma+1}}, \quad \text{with } q = 6R \left( 2\|C\| + \frac{M\kappa}{\sqrt{\|C\|}} + \sqrt{\frac{Q\sigma^2}{\|C\|^\gamma}} \right) \log \frac{6}{\delta}, \quad (9)$$

*with $\hat{f}_{\lambda,m}$ as in (5), $\lambda = \|C\| n^{-\frac{1}{2v+\gamma+1}}$ and*

1. *for plain Nyström*

$$m \geq (67 \vee 5\mathcal{N}_\infty(\lambda)) \log \frac{12\kappa^2}{\lambda\delta};$$

2. *for ALS Nyström and $T$-approximate leverage scores with subsampling probabilities $P_\lambda$, $t_0 \geq \frac{19\kappa^2}{n} \log \frac{12n}{\delta}$ and*

$$m \geq (334 \vee 78T^2\mathcal{N}(\lambda)) \log \frac{48n}{\delta}.$$

We add several comments. First, the above results can be shown to be optimal in a minimax sense. Indeed, minimax lower bounds proved in [26, 28] show that the learning rate in (9) is optimal under the considered assumptions (see Thm. 2, 3 of [26], for a discussion on minimax lower bounds see Sec. 2 of [26]). Second, the obtained bounds can be compared to those obtained for other regularized learning techniques. Techniques known to achieve optimal error rates include Tikhonov regularization [26, 28, 29], iterative regularization by early stopping [30, 31], spectral cut-off regularization (a.k.a. principal component regression or truncated SVD) [30, 31], as well as regularized stochastic gradient methods [32]. All these techniques are essentially equivalent from a statistical point of view and differ only in the required computations. For example, iterative methods allow for a computation of solutions corresponding to different regularization levels which is more efficient than Tikhonov or SVD based approaches. The key observation is that all these methods have the same $O(n^2)$ memory requirement. In this view, our results show that randomized subsampling methods can break such a memory barrier, and consequently achieve much better time complexity, while preserving optimal learning guarantees. Finally, we can compare our results with previous analysis of randomized kernel methods. As already mentioned, results close to those in Theorem 1 are given in [23, 24] in a fixed design setting. Our results extend and generalize the conclusions of these papers to a general statistical learning setting. Relevant results are given in [8] for a different approach, based on averaging KRLS solutions obtained splitting the data in $m$ groups (*divide and conquer RLS*). The analysis in [8] is only in expectation, but considers random design and shows that the proposed method is indeed optimal provided the number of splits is chosen depending on the effective dimension $\mathcal{N}(\lambda)$. This is the only other work we are aware of establishing optimal learning rates for randomized kernel approaches in a statistical learning setting. In comparison with Nyström computational regularization the main disadvantage of the divide and conquer approach is computational and in the model selection phase where solutions corresponding to different regularization parameters and number of splits usually need to be computed.

The proof of Theorem 1 is fairly technical and lengthy. It incorporates ideas from [26] and techniques developed to study spectral filtering regularization [30, 33]. In the next section, we briefly sketch some main ideas and discuss how they suggest an interesting perspective on regularization techniques including subsampling.

### 3.2 Proof sketch and a computational regularization perspective

A key step in the proof of Theorem 1 is an error decomposition, and corresponding bound, for any fixed $\lambda$ and $m$. Indeed, it is proved in Theorem 2 and Proposition 2 that, for $\delta > 0$, with probability

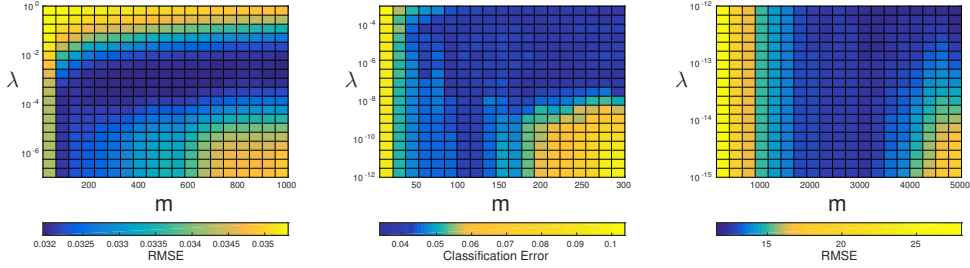

Figure 1: Validation errors associated to $20 \times 20$ grids of values for $m$ (x axis) and $\lambda$ (y axis) on `pumadyn32nh` (left), `breast cancer` (center) and `cpuSmall` (right).

at least $1 - \delta$,

$$\left| \mathcal{E}(\hat{f}_{\lambda,m}) - \mathcal{E}(f_{\mathcal{H}}) \right|^{1/2} \lesssim R \left( \frac{M\sqrt{\mathcal{N}_\infty(\lambda)}}{n} + \sqrt{\frac{\sigma^2 \mathcal{N}(\lambda)}{n}} \right) \log \frac{6}{\delta} + R\mathcal{C}(m)^{1/2+v} + R\lambda^{1/2+v}.$$

(10)

The first and last term in the right hand side of the above inequality can be seen as forms of *sample and approximation errors* [25] and are studied in Lemma 4 and Theorem 2. The mid term can be seen as a *computational error* and depends on the considered subsampling scheme. Indeed, it is shown in Proposition 2 that $\mathcal{C}(m)$ can be taken as,

$$\mathcal{C}_{\mathrm{pl}}(m) = \min \left\{ t > 0 \;\middle|\; (67 \vee 5\mathcal{N}_\infty(t)) \log \frac{12\kappa^2}{t\delta} \leq m \right\},$$

for the plain Nyström approach, and

$$\mathcal{C}_{\mathrm{ALS}}(m) = \min \left\{ \frac{19\kappa^2}{n} \log \frac{12n}{\delta} \leq t \leq \|C\| \;\middle|\; 78T^2\mathcal{N}(t) \log \frac{48n}{\delta} \leq m \right\},$$

for the approximate leverage scores approach. The bounds in Theorem 1 follow by: 1) minimizing in $\lambda$ the sum of the first and third term 2) choosing $m$ so that the computational error is of the same order of the other terms. Computational resources and regularization are then tailored to the generalization properties of the data at hand. We add a few comments. First, note that the error bound in (10) holds for a large class of subsampling schemes, as discussed in Section C.1 in the appendix. Then specific error bounds can be derived developing computational error estimates. Second, the error bounds in Theorem 2 and Proposition 2, and hence in Theorem 1, easily generalize to a larger class of regularization schemes beyond Tikhonov approaches, namely spectral filtering [30]. For space constraints, these extensions are deferred to a longer version of the paper. Third, we note that, in practice, optimal data driven parameter choices, e.g. based on hold-out estimates [31], can be used to adaptively achieve optimal learning bounds.

Finally, we observe that a different perspective is derived starting from inequality (10), and noting that the role played by $m$ and $\lambda$ can also be exchanged. Letting $m$ play the role of a regularization parameter, $\lambda$ can be set as a function of $m$ and $m$ tuned adaptively. For example, in the case of a plain Nyström approach, if we set

$$\lambda = \frac{\log m}{m}, \quad \text{and} \quad m = 3n^{\frac{1}{2v+\gamma+1}} \log n,$$

then the obtained learning solution achieves the error bound in Eq. (9). As above, the subsampling level can also be chosen by cross-validation. Interestingly, in this case by tuning $m$ we naturally control computational resources and regularization. An advantage of this latter parameterization is that, as described in the following, the solution corresponding to different subsampling levels is easy to update using Cholesky rank-one update formulas [34]. As discussed in the next section, in practice, a joint tuning over $m$ and $\lambda$ can be done starting from small $m$ and appears to be advantageous both for error and computational performances.

## 4 Incremental updates and experimental analysis

In this section, we first describe an incremental strategy to efficiently explore different subsampling levels and then perform extensive empirical tests aimed in particular at: 1) investigating the statistical and computational benefits of considering varying subsampling levels, and 2) compare the

**Input:** Dataset $(x_i, y_i)_{i=1}^n$, Subsampling $(\tilde{x}_j)_{j=1}^m$,
Regularization Parameter $\lambda$.
**Output:** Nyström KRLS estimators $\{\tilde{\alpha}_1, \dots, \tilde{\alpha}_m\}$.
Compute $\gamma_1$; $R_1 \leftarrow \sqrt{\gamma_1}$;
**for** $t \in \{2, \dots, m\}$ **do**
    Compute $A_t, u_t, v_t$;
    $R_t \leftarrow \begin{pmatrix} R_{t-1} & 0 \\ 0 & 0 \end{pmatrix}$;  $R_t \leftarrow \texttt{cholup}(R_t, u_t,' +')$;
    $\phantom{R_t \leftarrow} \begin{pmatrix} \phantom{R} & \phantom{0} \\ \phantom{0} & \phantom{0} \end{pmatrix}$  $R_t \leftarrow \texttt{cholup}(R_t, v_t,' -')$;
    $\tilde{\alpha}_t \leftarrow R_t^{-1}(R_t^{-\top}(A_t y))$;
**end for**

Algorithm 1: Incremental Nyström KRLS.

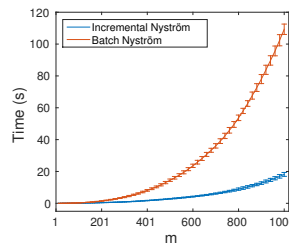

Figure 2: Model selection time on the `cpuSmall` dataset. $m \in [1, 1000]$ and $T = 50$, 10 repetitions.

performance of the algorithm with respect to state of the art solutions on several large scale benchmark datasets. Throughout this section, we only consider a plain Nyström approach, deferring to future work the analysis of leverage scores based sampling techniques. Interestingly, we will see that such a basic approach can often provide state of the art performances.

## 4.1 Efficient incremental updates

Algorithm 1 efficiently compute solutions corresponding to different subsampling levels, by exploiting rank-one Cholesky updates [34]. The proposed procedure allows to efficiently compute a whole regularization path of solutions, and hence perform fast model selection[2] (see Sect. A). In Algorithm 1, the function `cholup` is the Cholesky rank-one update formula available in many linear algebra libraries. The total cost of the algorithm is $O(nm^2 + m^3)$ time to compute $\tilde{\alpha}_2, \dots, \tilde{\alpha}_m$, while a naive non-incremental algorithm would require $O(nm^2 T + m^3 T)$ with $T$ is the number of analyzed subsampling levels. The following are some quantities needed by the algorithm: $A_1 = a_1$ and $A_t = (A_{t-1} \; a_t) \in \mathbb{R}^{n \times t}$, for any $2 \le t \le m$. Moreover, for any $1 \le t \le m$, $g_t = \sqrt{1 + \gamma_t}$ and

$$u_t = (c_t/(1 + g_t), \; g_t), \quad a_t = (K(\tilde{x}_t, x_1), \dots, K(\tilde{x}_t, x_n)), \quad c_t = A_{t-1}^\top a_t + \lambda n b_t,$$
$$v_t = (c_t/(1 + g_t), \; -1), \quad b_t = (K(\tilde{x}_t, \tilde{x}_1), \dots, K(\tilde{x}_t, \tilde{x}_{t-1})), \quad \gamma_t = a_t^\top a_t + \lambda n K(\tilde{x}_t, \tilde{x}_t).$$

## 4.2 Experimental analysis

We empirically study the properties of Algorithm 1, considering a Gaussian kernel of width $\sigma$. The selected datasets are already divided in a training and a test part[3]. We randomly split the training part in a training set and a validation set (80% and 20% of the $n$ training points, respectively) for parameter tuning via cross-validation. The $m$ subsampled points for Nyström approximation are selected uniformly at random from the training set. We report the performance of the selected model on the fixed test set, repeating the process for several trials.

**Interplay between $\lambda$ and $m$.** We begin with a set of results showing that incrementally exploring different subsampling levels can yield very good performance while substantially reducing the computational requirements. We consider the `pumadyn32nh` ($n = 8192$, $d = 32$), the `breast cancer` ($n = 569$, $d = 30$), and the `cpuSmall` ($n = 8192$, $d = 12$) datasets[4]. In Figure 1, we report the validation errors associated to a $20 \times 20$ grid of values for $\lambda$ and $m$. The $\lambda$ values are logarithmically spaced, while the $m$ values are linearly spaced. The ranges and kernel bandwidths, chosen according to preliminary tests on the data, are $\sigma = 2.66$, $\lambda \in [10^{-7}, 1]$, $m \in [10, 1000]$ for `pumadyn32nh`, $\sigma = 0.9$, $\lambda \in [10^{-12}, 10^{-3}]$, $m \in [5, 300]$ for `breast cancer`, and $\sigma = 0.1$, $\lambda \in [10^{-15}, 10^{-12}]$, $m \in [100, 5000]$ for `cpuSmall`. The main observation that can be derived from this first series of tests is that a small $m$ is sufficient to obtain the same results achieved with the largest $m$. For example, for `pumadyn32nh` it is sufficient to choose $m = 62$ and $\lambda = 10^{-7}$ to obtain an average test RMSE of 0.33 over 10 trials, which is the same as the one obtained using $m = 1000$ and $\lambda = 10^{-3}$, with a 3-fold speedup of the joint training and validation phase. Also, it is interesting to observe that for given values of $\lambda$, large values of $m$ can decrease the performance. This observation is consistent with the results in Section 3.1, showing that $m$ can play the

Table 1: Test RMSE comparison for exact and approximated kernel methods. The results for KRLS, Batch Nyström, RF and Fastfood are the ones reported in [6]. $n_{tr}$ is the size of the training set.

| Dataset | $n_{tr}$ | $d$ | Incremental Nyström RBF | KRLS RBF | Batch Nyström RBF | RF RBF | Fastfood RBF | Fastfood FFT | KRLS Matern | Fastfood Matern |
|---|---|---|---|---|---|---|---|---|---|---|
| Insurance Company | 5822 | 85 | $0.23180 \pm 4 \times 10^{-5}$ | **0.231** | 0.232 | 0.266 | 0.264 | 0.266 | 0.234 | 0.235 |
| CPU | 6554 | 21 | $\mathbf{2.8466 \pm 0.0497}$ | 7.271 | 6.758 | 7.103 | 7.366 | 4.544 | 4.345 | 4.211 |
| CT slices (axial) | 42800 | 384 | $\mathbf{7.1106 \pm 0.0772}$ | NA | 60.683 | 49.491 | 43.858 | 58.425 | NA | 14.868 |
| Year Prediction MSD | 463715 | 90 | $\mathbf{0.10470 \pm 5 \times 10^{-5}}$ | NA | 0.113 | 0.123 | 0.115 | 0.106 | NA | 0.116 |
| Forest | 522910 | 54 | $0.9638 \pm 0.0186$ | NA | **0.837** | 0.840 | 0.840 | 0.838 | NA | 0.976 |

role of a regularization parameter. Similar results are obtained for `breast cancer`, where for $\lambda = 4.28 \times 10^{-6}$ and $m = 300$ we obtain a $1.24\%$ average classification error on the test set over 20 trials, while for $\lambda = 10^{-12}$ and $m = 67$ we obtain $1.86\%$. For `cpuSmall`, with $m = 5000$ and $\lambda = 10^{-12}$ the average test RMSE over 5 trials is 12.2, while for $m = 2679$ and $\lambda = 10^{-15}$ it is only slightly higher, 13.3, but computing its associated solution requires less than half of the time and approximately half of the memory.

**Regularization path computation**. If the subsampling level $m$ is used as a regularization parameter, the computation of a regularization path corresponding to different subsampling levels becomes crucial during the model selection phase. A naive approach, that consists in recomputing the solutions of Eq. 5 for each subsampling level, would require $O(m^2 nT + m^3 LT)$ computational time, where $T$ is the number of solutions with different subsampling levels to be evaluated and $L$ is the number of Tikhonov regularization parameters. On the other hand, by using the incremental Nyström algorithm the model selection time complexity is $O(m^2 n + m^3 L)$ for the whole regularization path. We experimentally verify this speedup on `cpuSmall` with 10 repetitions, setting $m \in [1, 5000]$ and $T = 50$. The model selection times, measured on a server with $12 \times 2.10$GHz Intel® Xeon® E5-2620 v2 CPUs and 132 GB of RAM, are reported in Figure 2. The result clearly confirms the beneficial effects of incremental Nyström model selection on the computational time.

**Predictive performance comparison**. Finally, we consider the performance of the algorithm on several large scale benchmark datasets considered in [6], see Table 1. $\sigma$ has been chosen on the basis of preliminary data analysis. $m$ and $\lambda$ have been chosen by cross-validation, starting from small subsampling values up to $m_{max} = 2048$, and considering $\lambda \in [10^{-12}, 1]$. After model selection, we retrain the best model on the entire training set and compute the RMSE on the test set. We consider 10 trials, reporting the performance mean and standard deviation. The results in Table 1 compare Nyström computational regularization with the following methods (as in [6]):

- **Kernel Regularized Least Squares (KRLS):** Not compatible with large datasets.
- **Random Fourier features (RF):** As in [4], with a number of random features $D = 2048$.
- **Fastfood RBF, FFT and Matern kernel:** As in [6], with $D = 2048$ random features.
- **Batch Nyström:** Nyström method [3] with uniform sampling and $m = 2048$.

The above results show that the proposed incremental Nyström approach behaves really well, matching state of the art predictive performances.

### Acknowledgments

The work described in this paper is supported by the Center for Brains, Minds and Machines (CBMM), funded by NSF STC award CCF-1231216; and by FIRB project RBFR12M3AC, funded by the Italian Ministry of Education, University and Research.

## Footnotes

[1] If $\mathcal{N}_\infty(\lambda)$ is finite, then $\mathcal{N}_\infty(\|C\|) = \sup_{x \in X} \|(C + \|C\|I)^{-1}K_x\|^2 \geq 1/2\|C\|^{-1}\sup_{x \in X} \|K_x\|^2$, therefore $K(x, x) \leq 2\|C\|\mathcal{N}_\infty(\|C\|)$.

[2]The code for Algorithm 1 is available at `lcsl.github.io/NystromCoRe`.

[3]In the following we denote by $n$ the total number of points and by $d$ the number of dimensions.

[4]`www.cs.toronto.edu/~delve` and `archive.ics.uci.edu/ml/datasets`

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
