[Supplementary Material · appendix.pdf]

# A   The incremental algorithm

Let $(x_i, y_i)_{i=1}^n$ be the dataset and $(\tilde{x}_i)_{i=1}^m$ be the selected Nyström points. We want to compute $\tilde{\alpha}$ of Eq. 5, incrementally in $m$. Towards this goal we compute an incremental Cholesky decomposition $R_t$ for $t \in \{1, \ldots, m\}$ of the matrix $G_t = K_{nt}^\top K_{nt} + \lambda n K_{tt}$, and the coefficients $\tilde{\alpha}_t$ by $\tilde{\alpha}_t = R_t^{-1} R_t^{-\top} K_{nt}^\top y$. Note that, for any $1 \le t \le m-1$, by assuming $G_t = R_t^\top R_t$ for an upper triangular matrix $R_t$, we have

$$G_{t+1} = \begin{pmatrix} G_t & c_{t+1} \\ c_{t+1}^\top & \gamma_{t+1} \end{pmatrix} = \begin{pmatrix} R_t & 0 \\ 0 & 0 \end{pmatrix}^\top \begin{pmatrix} R_t & 0 \\ 0 & 0 \end{pmatrix} + C_{t+1} \quad \text{with} \quad C_{t+1} = \begin{pmatrix} 0 & c_{t+1} \\ c_{t+1}^\top & \gamma_{t+1} \end{pmatrix},$$

and $c_{t+1}, \gamma_{t+1}$ as in Section 4.1. Note moreover that $G_1 = \gamma_1$. Thus if we decompose the matrix $C_{t+1}$ in the form $C_{t+1} = u_{t+1} u_{t+1}^\top - v_{t+1} v_{t+1}^\top$ we are able compute $R_{t+1}$, the Cholesky matrix of $G_{t+1}$, by updating a bordered version of $R_t$ with two rank-one Cholesky updates. This is exactly Algorithm 1 with $u_{t+1}$ and $v_{t+1}$ as in Section 4.1. Note that the rank-one Cholesky update requires $O(t^2)$ at each call, while the computation of $c_t$ requires $O(nt)$ and the ones of $\tilde{\alpha}_t$ requires to solve two triangular linear systems, that is $O(t^2 + nt)$. Therefore the total cost for computing $\tilde{\alpha}_2, \ldots, \tilde{\alpha}_m$ is $O(nm^2 + m^3)$.

# B   Preliminary definitions

We begin introducing several operators that will be useful in the following. Let $z_1, \ldots, z_m \in \mathcal{H}$ and for all $f \in \mathcal{H}$, $a \in \mathbb{R}^m$, let

$$Z_m : \mathcal{H} \to \mathbb{R}^m, \qquad Z_m f = (\langle z_1, f \rangle_{\mathcal{H}}, \ldots, \langle z_m, f \rangle_{\mathcal{H}}),$$
$$Z_m^* : \mathbb{R}^m \to \mathcal{H}, \qquad Z_m^* a = \sum_{i=1}^m a_i z_i.$$

Let $S_n = \frac{1}{\sqrt{n}} Z_m$ and $S_n^* = \frac{1}{\sqrt{n}} Z_m^*$ the operators obtained taking $m = n$ and $z_i = K_{x_i}$, $\forall i = 1, \ldots, n$ in the above definitions. Moreover, for all $f, g \in \mathcal{H}$ let

$$C_n : \mathcal{H} \to \mathcal{H}, \quad \langle f, C_n g \rangle_{\mathcal{H}} = \frac{1}{n} \sum_{i=1}^n f(x_i) g(x_i).$$

The above operators are linear and finite rank. Moreover $C_n = S_n^* S_n$ and $K_n = n S_n S_n^*$, and further $B_{nm} = \sqrt{n} S_n Z_m^* \in \mathbb{R}^{n \times m}$, $G_{mm} = Z_m Z_m^* \in \mathbb{R}^{m \times m}$ and $\tilde{K}_n = B_{nm} G_{mm}^\dagger B_{nm}^\top \in \mathbb{R}^{n \times n}$.

# C   Representer theorem for Nyström computational regularization and extensions

In this section we consider explicit representations of the estimator obtained via Nyström computational regularization and extensions. Indeed, we consider a general subspace $\mathcal{H}_m$ of $\mathcal{H}$, and the following problem

$$\hat{f}_{\lambda,m} = \underset{f \in \mathcal{H}_m}{\operatorname{argmin}} \frac{1}{n} \sum_{i=1}^n (f(x_i) - y_i)^2 + \lambda \|f\|_{\mathcal{H}}^2. \tag{11}$$

In the following lemmas, we show three different characterizations of $f_{\lambda,m}$.

**Lemma 1.** *Let $f_{\lambda,m}$ be the solution of the problem in Eq. (11). Then it is characterized by the following equation*

$$(P_m C_n P_m + \lambda I)\hat{f}_{\lambda,m} = P_m S_n^* \hat{y}_n, \tag{12}$$

*with $P_m$ the projection operator with range $\mathcal{H}_m$ and $\hat{y}_n = \frac{1}{\sqrt{n}} y$.*

*Proof.* The proof proceeds in three steps. First, note that, by rewriting Problem (11) with the notation introduced in the previous section, we obtain,

$$\hat{f}_{\lambda,m} = \underset{f \in \mathcal{H}_m}{\operatorname{argmin}} \|S_n f - \hat{y}_n\|^2 + \lambda \|f\|_{\mathcal{H}}^2. \tag{13}$$

This problem is strictly convex and coercive, therefore admits a unique solution. Second, we show that its solution coincide to the one of the following problem,

$$\hat{f}^* = \underset{f \in \mathcal{H}}{\operatorname{argmin}} \|S_n P_m f - \widehat{y}_n\|^2 + \lambda \|f\|_{\mathcal{H}}^2. \tag{14}$$

Note that the above problem is again strictly convex and coercive. To show that $\hat{f}_{\lambda,m} = \hat{f}^*$, let $\hat{f}^* = a + b$ with $a \in \mathcal{H}_m$ and $b \in \mathcal{H}_m^\perp$. A necessary condition for $\hat{f}^*$ to be optimal, is that $b = 0$, indeed, considering that $P_m b = 0$, we have

$$\|S_n P_m f^* - \widehat{y}_n\|^2 + \lambda \|f^*\|_{\mathcal{H}}^2 = \|S_n P_m a - \widehat{y}_n\|^2 + \lambda \|a\|_{\mathcal{H}}^2 + \lambda \|b\|_{\mathcal{H}}^2 \geq \|S_n P_m a - \widehat{y}_n\|^2 + \lambda \|a\|_{\mathcal{H}}^2.$$

This means that $\hat{f}^* \in \mathcal{H}_m$, but on $\mathcal{H}_m$ the functionals defining Problem (13) and Problem (14) are identical because $P_m f = f$ for any $f \in \mathcal{H}_m$ and so $\hat{f}_{\lambda,m} = \hat{f}^*$. Therefore, by computing the derivative of the functional of Problem (14), we see that $\hat{f}_{\lambda,m}$ is given by Eq. (12). $\qquad \square$

Using the above results, we can give an equivalent representations of the function $\hat{f}_{\lambda,m}$. Towards this end, let $Z_m$ be a linear operator as in Sect. B such that the range of $Z_m^*$ is exactly $\mathcal{H}_m$. Morever, let

$$Z_m = U \Sigma V^*$$

be the SVD of $Z_m$ where $U : \mathbb{R}^t \to \mathbb{R}^m$, $\Sigma : \mathbb{R}^t \to \mathbb{R}^t$, $V : \mathbb{R}^t \to \mathcal{H}$, $t \leq m$ and $\Sigma = \operatorname{diag}(\sigma_1, \ldots, \sigma_t)$ with $\sigma_1 \geq \cdots \geq \sigma_t > 0$, $U^* U = I_t$ and $V^* V = I_t$. Then the orthogonal projection operator $P_m$ is given by $P_m = V V^*$ and the range of $P_m$ is exactly $\mathcal{H}_m$. In the following lemma we give a characterization of $\hat{f}_{\lambda,m}$ that will be useful in the proof of the main theorem.

**Lemma 2.** *Given the above definitions, $\hat{f}_{\lambda,m}$ can be written as*

$$\hat{f}_{\lambda,m} = V(V^* C_n V + \lambda I)^{-1} V^* S_n^* \widehat{y}_n. \tag{15}$$

*Proof.* By Lemma 1, we know that $\hat{f}_{\lambda,m}$ is written as in Eq. (12). Now, note that $\hat{f}_{\lambda,m} = P_m \hat{f}_{\lambda,m}$ and Eq. (12) imply $(P_m C_m P_m + \lambda I) P_m \hat{f}_{\lambda,m} = P_m S_n^* \widehat{y}_n$, that is equivalent to

$$V(V^* C_n V + \lambda I) V^* \hat{f}_{\lambda,m} = V V^* S_n^* \widehat{y}_n,$$

by substituting $P_m$ with $V V^*$. Thus by premultiplying the previous equation by $V^*$ and dividing by $V^* C_m V + \lambda I$, we have

$$V^* \hat{f}_{\lambda,m} = (V^* C_m V + \lambda I)^{-1} V^* S_n^* \widehat{y}_n.$$

Finally, by premultiplying by $V$,

$$\hat{f}_{\lambda,m} = P_m \hat{f}_{\lambda,m} = V(V^* C_m V + \lambda I)^{-1} V^* S_n^* \widehat{y}_n.$$

$\qquad \square$

Finally, the following result provide a characterization of the solution useful for computations.

**Lemma 3** (Representer theorem for $\hat{f}_{\lambda,m}$)**.** *Given the above definitions, we have that $\hat{f}_{\lambda,m}$ can be written as*

$$\hat{f}_{\lambda,m}(x) = \sum_{i=1}^m \tilde{\alpha}_i z_i(x), \quad \text{with } \tilde{\alpha} = (B_{nm}^\top B_{nm} + \lambda n G_{mm})^\dagger B_{nm}^\top y \quad \forall \, x \in X. \tag{16}$$

*Proof.* According to the definitions of $B_{nm}$ and $G_{mm}$ we have that

$$\tilde{\alpha} = (B_{nm}^\top B_{nm} + \lambda n G_{mm})^\dagger B_{nm}^\top y = ((Z_m S_n^*)(S_n Z_m^*) + \lambda (Z_m Z_m^*))^\dagger (Z_m S_n^*) \widehat{y}_n.$$

Moreover, according to the definition of $Z_m$ we have

$$\hat{f}_{\lambda,m}(x) = \sum_{i=1}^m \tilde{\alpha}_i \langle z_i, K_x \rangle = \langle Z_m K_x, \tilde{\alpha} \rangle_{\mathbb{R}^m} = \langle K_x, Z_m^* \tilde{\alpha} \rangle_{\mathcal{H}} \quad \forall \, x \in X,$$

so that

$$\hat{f}_{\lambda,m} = Z_m^*((Z_m S_n^*)(S_n Z_m^*) + \lambda(Z_m Z_m^*))^\dagger (Z_m S_n^*)\widehat{y}_n = Z_m^*(Z_m C_{n\lambda} Z_m^*)^\dagger (Z_m S_n^*)\widehat{y}_n,$$

where $C_{n\lambda} = C_n + \lambda I$. Let $F = U\Sigma$, $G = V^* C_n V + \lambda I$, $H = \Sigma U^\top$, and note that $F$, $GH$, $G$ and $H$ are full-rank matrices, then we can perform the full-rank factorization of the pseudo-inverse (see Eq.24, Thm. 5, Chap. 1 of [1]) obtaining

$$(Z_m C_{n\lambda} Z_m^*)^\dagger = (FGH)^\dagger = H^\dagger (FG)^\dagger = H^\dagger G^{-1} F^\dagger = U\Sigma^{-1}(V^* C_n V + \lambda I)^{-1}\Sigma^{-1} U^*.$$

Finally, simplyfing $U$ and $\Sigma$, we have

$$\begin{aligned}
\hat{f}_{\lambda,m} &= Z_m^*(Z_m C_{n\lambda} Z_m^*)^\dagger (Z_m S_n^*)\widehat{y}_n \\
&= V\Sigma U^* U\Sigma^{-1}(V^* C_n V + \lambda I)^{-1}\Sigma^{-1} U^* U\Sigma V^* S_n^* \widehat{y}_n \\
&= V(V^* C_n V + \lambda I)^{-1} V^* S_n^* \widehat{y}_n.
\end{aligned}$$

$\square$

## C.1    Extensions

Inspection of the proof shows that our analysis extends beyond the class of subsampling schemes in Theorem 1. Indeed, the error decomposition Theorem 2 directly applies to a large family of approximation schemes. Several further examples are described next.

**KRLS and Generalized Nyström**    In general we could choose an arbitrary $\mathcal{H}_m \subseteq \mathcal{H}$. Let $Z_m : \mathcal{H} \to \mathbb{R}^m$ be a linear operator such that

$$\mathcal{H}_m = \operatorname{ran} Z_m^* = \{f \mid f = Z_m^* \alpha, \ \alpha \in \mathbb{R}^m\}. \tag{17}$$

Without loss of generality, $Z_m^*$ is expressible as $Z_m^* = (z_1, \ldots, z_m)^\top$ with $z_1, \ldots, z_m \in \mathcal{H}$, therefore, according to Section B and to Lemma 3, the solution of KRLS approximated with the generalized Nyström scheme is

$$\hat{f}_{\lambda,m}(x) = \sum_{i=1}^m \tilde{\alpha}_i z_i(x), \quad \text{with } \tilde{\alpha} = (B_{nm}^\top B_{nm} + \lambda n G_{mm})^\dagger B_{nm}^\top y \tag{18}$$

with $B_{nm} \in \mathbb{R}^{n \times m}$, $(B_{nm})_{ij} = z_j(x_i)$ and $G_{mm} \in \mathbb{R}^{m \times m}$, $(G_{mm})_{ij} = \langle z_i, z_j \rangle_{\mathcal{H}}$, or equivalently

$$\hat{f}_{\lambda,m}(x) = \sum_{i=1}^m \tilde{\alpha}_i z_i(x), \quad \tilde{\alpha} = G_{mm}^\dagger B_{nm}^\top (\tilde{K}_n + \lambda n I)^\dagger \widehat{y}_n, \quad \tilde{K}_n = B_{nm} G_{mm}^\dagger B_{nm}^\top \tag{19}$$

The following are some examples of Generalized Nyström approximations.

**Plain Nyström with various sampling schemes [2–4]**    For a realization $s : \mathbb{N} \to \{1, \ldots, n\}$ of a given sampling scheme, we choose $Z_m = S_m$ with $S_m^* = (K_{x_{s(1)}}, \ldots, K_{x_{s(m)}})^\top$ where $(x_i)_{i=1}^n$ is the training set. With such $Z_m$ we obtain $\tilde{K}_n = K_{nm}(K_{mm})^\dagger K_{nm}^\top$ and so Eq. (18) becomes exactly Eq. (5).

**Reduced rank Plain Nyström [5]**    Let $p \geq m$, $S_p$ as in the previous example, the linear operator associated to $p$ points of the dataset. Let $K_{pp} = S_p S_p^\top \in \mathbb{R}^{p \times p}$, that is $(K_{pp})_{ij} = K(x_i, x_j)$. Let $K_{pp} = \sum_{i=1}^p \sigma_i u_i u_i^\top$ its eigenvalue decomposition and $U_m = (u_1, \ldots, u_m)$. Let $(K_{pp})_m = U_m^\top K_{pp} U_m$ be the $m$-rank approximation of $K_{pp}$. We approximate this family by choosing $Z_m = U_m^\top S_p$, indeed we obtain $\tilde{K}_n = K_{nm} U_m (U_m^\top K_{pp} U_m)^\dagger U_m^\top K_{nm}^\top = K_{nm}(K_{pp})_m^\dagger K_{nm}^\top$.

**Nyström with sketching matrices [6]**    We cover this family by choosing $Z_m = R_m S_n$, where $S_n$ is the same operator as in the plain Nyström case where we select all the points of the training set and $R_m$ a $m \times n$ sketching matrix. In this way we have $\tilde{K}_n = K_n R_m^*(R_m K_n R_m^*)^\dagger R_m K_n$, that is exactly the SPSD sketching model.

## D  Probabilistic inequalities

In this section we collect five main probabilistic inequalities needed in the proof of the main result. We let $\rho_X$ denote the marginal distribution of $\rho$ on $X$ and $\rho(\cdot|x)$ the conditional distribution on $\mathbb{R}$ given $x \in X$. Lemmas 6, 7 and especially Proposition 1 are new and of interest in their own right.

The first result is essentially taken from [7].

**Lemma 4** (Sample Error). *Under Assumptions 1, 2 and 3, for any $\delta > 0$, the following holds with probability $1 - \delta$*

$$\|(C + \lambda I)^{-1/2}(S_n^* \widehat{y}_n - C_n f_{\mathcal{H}})\| \leq 2 \left( \frac{M \sqrt{\mathcal{N}_\infty(\lambda)}}{n} + \sqrt{\frac{\sigma^2 \mathcal{N}(\lambda)}{n}} \right) \log \frac{2}{\delta}.$$

*Proof.* The proof is given in [7] for bounded kernels and the slightly stronger condition $\int (e^{\frac{|y - f_{\mathcal{H}}(x)|}{M}} - \frac{|y - f_{\mathcal{H}}(x)|}{M} - 1)d\rho(y|x) \leq \sigma^2/M^2$ in place of Assumption 2. More precisely, note that

$$(C + \lambda I)^{-1/2}(S_n^* \widehat{y}_n - C_n f_{\mathcal{H}}) = \frac{1}{n} \sum_{i=1}^n \zeta_i,$$

where $\zeta_1, \ldots, \zeta_n$ are i.i.d. random variables, defined as $\zeta_i = (C + \lambda I)^{-1/2} K_{x_i}(y_i - f_{\mathcal{H}}(x_i))$. For any $1 \leq i \leq n$,

$$\mathbb{E}\zeta_i = \int_{X \times \mathbb{R}} (C + \lambda I)^{-1/2} K_{x_i}(y_i - f_{\mathcal{H}}(x_i))d\rho(x_i, y_i)$$

$$= \int_X (C + \lambda I)^{-1/2} K_{x_i} \int_{\mathbb{R}} (y_i - f_{\mathcal{H}}(x_i))d\rho(y_i|x_i)d\rho_X(x_i) = 0,$$

almost everywhere by Assumption 1 (see Step 3.2 of Thm. 4 in [7]). In the same way we have

$$\mathbb{E}\|\zeta_i\|^p = \int_{X \times \mathbb{R}} \|(C + \lambda I)^{-1/2} K_{x_i}(y_i - f_{\mathcal{H}}(x_i))\|^p d\rho(x_i, y_i)$$

$$= \int_X \|(C + \lambda I)^{-1/2} K_{x_i}\|^p \int_{\mathbb{R}} |y_i - f_{\mathcal{H}}(x_i)|^p d\rho(y_i|x_i)d\rho_X(x_i)$$

$$\leq \sup_{x \in X} \|(C + \lambda I)^{-1/2} K_x\|^{p-2} \int_X \|(C + \lambda I)^{-1/2} K_{x_i}\|^2 \int_{\mathbb{R}} |y_i - f_{\mathcal{H}}(x_i)|^p d\rho(y_i|x_i)d\rho_X(x_i)$$

$$\leq \frac{1}{2} p! \sqrt{\sigma^2 \mathcal{N}(\lambda)}^2 (M \sqrt{\mathcal{N}_\infty(\lambda)})^{p-2},$$

where $\sup_{x \in X} \|(C + \lambda I)^{-1/2} K_x\| = \sqrt{\mathcal{N}_\infty(\lambda)}$ and $\int_X \|(C + \lambda I)^{-1/2} K_{x_i}\|^2 = \mathcal{N}(\lambda)$ by Assumption 3, while the bound on the moments of $y - f(x)$ is given in Assumption 2. Finally, to concentrate the sum of random vectors, we apply Prop. 11. $\square$

The next result is taken from [8].

**Lemma 5.** *Under Assumption 3, for any $\delta \geq 0$ and $\frac{9\kappa^2}{n} \log \frac{n}{\delta} \leq \lambda \leq \|C\|$, the following inequality holds with probability at least $1 - \delta$,*

$$\|(C_n + \lambda I)^{-1/2} C^{1/2}\| \leq \|(C_n + \lambda I)^{-1/2} (C + \lambda I)^{1/2}\| \leq 2.$$

*Proof.* Lemma 7 of [8] gives an the extended version of the above result. Our bound on $\lambda$ is scaled by $\kappa^2$ because in [8] it is assumed $\kappa \leq 1$. $\square$

**Lemma 6** (plain Nyström approximation). *Under Assumption 3, let $J$ be a partition of $\{1, \ldots, n\}$ chosen uniformly at random from the partitions of cardinality $m$. Let $\lambda > 0$, for any $\delta > 0$, such that $m \geq 67 \log \frac{4\kappa^2}{\lambda\delta} \vee 5\mathcal{N}_\infty(\lambda) \log \frac{4\kappa^2}{\lambda\delta}$, the following holds with probability $1 - \delta$*

$$\|(I - P_m)C^{1/2}\|^2 \leq 3\lambda,$$

*where $P_m$ is the projection operator on the subspace $\mathcal{H}_m = \text{span}\{K_{x_j} \mid j \in J\}$.*

*Proof.* Define the linear operator $C_m : \mathcal{H} \to \mathcal{H}$, as $C_m = \frac{1}{m}\sum_{j\in J} K_{x_j} \otimes K_{x_j}$. Now note that the range of $C_m$ is exactly $\mathcal{H}_m$. Therefore, by applying Prop. 3 and 7, we have that

$$\|(I - P_m)C_\lambda^{1/2}\|^2 \leq \lambda\|(C_m + \lambda I)^{-1/2}C^{1/2}\|^2 \leq \frac{\lambda}{1 - \beta(\lambda)},$$

with $\beta(\lambda) = \lambda_{\max}\left(C_\lambda^{-1/2}(C - C_m)C_\lambda^{-1/2}\right)$. To upperbound $\frac{\lambda}{1-\beta(\lambda)}$ we need an upperbound for $\beta(\lambda)$. Considering that, given the partition $J$, the random variables $\zeta_j = K_{x_j} \otimes K_{x_j}$ are i.i.d., then we can apply Prop. 8, to obtain

$$\beta(\lambda) \leq \frac{2w}{3m} + \sqrt{\frac{2w\mathcal{N}_\infty(\lambda)}{m}},$$

where $w = \log\frac{4\,\mathrm{Tr}(C)}{\lambda\delta}$ with probability $1 - \delta$. Thus, by choosing $m \geq 67w \vee 5\mathcal{N}_\infty(\lambda)w$, we have that $\beta(\lambda) \leq 2/3$, that is

$$\|(I - P_m)C_\lambda^{1/2}\|^2 \leq 3\lambda.$$

Finally, note that by definition $\mathrm{Tr}(C) \leq \kappa^2$. $\qquad\square$

**Lemma 7** (Nyström approximation for ALS selection method). *Let $(\hat{l}_i(t))_{i=1}^n$ be the collection of approximate leverage scores. Let $\lambda > 0$ and $P_\lambda$ be defined as $P_\lambda(i) = \hat{l}_i(\lambda)/\sum_{j\in N}\hat{l}_j(\lambda)$ for any $i \in N$ with $N = \{1,\dots,n\}$. Let $\mathfrak{I} = (i_1,\dots,i_m)$ be a collection of indices independently sampled with replacement from $N$ according to the probability distribution $P_\lambda$. Let $P_m$ be the projection operator on the subspace $\mathcal{H}_m = \mathrm{span}\{K_{x_j}|j \in J\}$ and $J$ be the subcollection of $\mathfrak{I}$ with all the duplicates removed. Under Assumption 3, for any $\delta > 0$ the following holds with probability $1 - 2\delta$*

$$\|(I - P_m)(C + \lambda I)^{1/2}\| \leq 3\lambda,$$

*when the following conditions are satisfied:*

1. *there exists a $T \geq 1$ and a $\lambda_0 > 0$ such that $(\hat{l}_i(t))_{i=1}^n$ are $T$-approximate leverage scores for any $t \geq \lambda_0$ (see Def. 1),*

2. $n \geq 1655\kappa^2 + 223\kappa^2 \log\frac{2\kappa^2}{\delta}$,

3. $\lambda_0 \vee \frac{19\kappa^2}{n}\log\frac{2n}{\delta} \leq \lambda \leq \|C\|$,

4. $m \geq 334\log\frac{8n}{\delta} \vee 78T^2\mathcal{N}(\lambda)\log\frac{8n}{\delta}$.

*Proof.* Define $\tau = \delta/4$. Next, define the diagonal matrix $H \in \mathbb{R}^{n\times n}$ with $(H)_{ii} = 0$ when $P_\lambda(i) = 0$ and $(H)_{ii} = \frac{nq(i)}{mP_\lambda(i)}$ when $P_\lambda(i) > 0$, where $q(i)$ is the number of times the index $i$ is present in the collection $\mathfrak{I}$. We have that

$$S_n^* H S_n = \frac{1}{m}\sum_{i=1}^n \frac{q(i)}{P_\lambda(i)}K_{x_i} \otimes K_{x_i} = \frac{1}{m}\sum_{j\in J}\frac{q(j)}{P_\lambda(j)}K_{x_j} \otimes K_{x_j}.$$

Now, considering that $\frac{q(j)}{P_\lambda(j)} > 0$ for any $j \in J$, thus $\mathrm{ran}\, S_n^* H S_n = \mathcal{H}_m$. Therefore, by using Prop. 3 and 7, we exploit the fact that the range of $P_m$ is the same of $S_n^* H S_n$, to obtain

$$\|(I - P_m)(C + \lambda I)^{1/2}\|^2 \leq \lambda\|(S_n^* H S_n + \lambda I)^{-1/2}C^{1/2}\|^2 \leq \frac{\lambda}{1 - \beta(\lambda)},$$

with $\beta(\lambda) = \lambda_{\max}\left(C_\lambda^{-1/2}(C - S_n^* H S_n)C_\lambda^{-1/2}\right)$. Considering that the function $(1 - x)^{-1}$ is increasing on $-\infty < x < 1$, in order to bound $\lambda/(1 - \beta(\lambda))$ we need an upperbound for $\beta(\lambda)$. Here we split $\beta(\lambda)$ in the following way,

$$\beta(\lambda) \leq \underbrace{\lambda_{\max}\left(C_\lambda^{-1/2}(C - C_n)C_\lambda^{-1/2}\right)}_{\beta_1(\lambda)} + \underbrace{\lambda_{\max}\left(C_\lambda^{-1/2}(C_n - S_n^* H S_n)C_\lambda^{-1/2}\right)}_{\beta_2(\lambda)}.$$

Considering that $C_n$ is the linear combination of independent random vectors, for the first term we can apply Prop. 8, obtaining a bound of the form

$$\beta_1(\lambda) \leq \frac{2w}{3n} + \sqrt{\frac{2w\kappa^2}{\lambda n}},$$

with probability $1 - \tau$, where $w = \log \frac{4\kappa^2}{\lambda \tau}$ (we used the fact that $\mathcal{N}_\infty(\lambda) \leq \kappa^2/\lambda$). Then, after dividing and multiplying by $C_{n\lambda}^{1/2}$, we split the second term $\beta_2(\lambda)$ as follows:

$$\begin{aligned}
\beta_2(\lambda) &\leq \|C_\lambda^{-1/2}(C_n - S_n^* H S_n)C_\lambda^{-1/2}\| \\
&\leq \|C_\lambda^{-1/2}C_{n\lambda}^{1/2}C_{n\lambda}^{-1/2}(C_n - S_n^* H S_n)C_{n\lambda}^{-1/2}C_{n\lambda}^{1/2}C_\lambda^{-1/2}\| \\
&\leq \|C_\lambda^{-1/2}C_{n\lambda}^{1/2}\|^2 \|C_{n\lambda}^{-1/2}(C_n - S_n^* H S_n)C_{n\lambda}^{-1/2}\|.
\end{aligned}$$

Let

$$\beta_3(\lambda) = \|C_{n\lambda}^{-1/2}(C_n - S_n^* H S_n)C_{n\lambda}^{-1/2}\| = \|C_{n\lambda}^{-1/2}S_n^*(I - H)S_n C_{n\lambda}^{-1/2}\|. \tag{20}$$

Note that $S_n C_{n\lambda}^{-1} S_n^* = K_n(K_n + \lambda n I)^{-1}$ indeed $C_{n\lambda}^{-1} = (S_n^* S_n + \lambda I)^{-1}$ and $K_n = n S_n S_n^*$. Therefore we have

$$S_n C_{n\lambda}^{-1} S_n^* = S_n(S_n^* S_n + \lambda I)^{-1}S_n^* = (S_n S_n^* + \lambda I)^{-1}S_n S_n^* = (K_n + \lambda n I)^{-1}K_n.$$

Thus, if we let $U\Sigma U^\top$ be the eigendecomposition of $K_n$, we have that $(K_n + \lambda n I)^{-1}K_n = U(\Sigma + \lambda n I)^{-1}\Sigma U^\top$ and thus $S_n C_{n\lambda}^{-1} S_n^* = U(\Sigma + \lambda n I)^{-1}\Sigma U^\top$. In particular this implies that $S_n C_{n\lambda}^{-1} S_n^* = U Q_n^{1/2} Q_n^{1/2} U^\top$ with $Q_n = (\Sigma + \lambda n I)^{-1}\Sigma$. Therefore we have

$$\beta_3(\lambda) = \|C_{n\lambda}^{-1/2}S_n^*(I - H)S_n C_{n\lambda}^{-1/2}\| = \|Q_n^{1/2}U^\top(I - H)U Q_n^{1/2}\|,$$

where we used twice the fact that $\|ABA^*\| = \|(A^*A)^{1/2}B(A^*A)^{1/2}\|$ for any bounded linear operators $A, B$.

Consider the matrix $A = Q_n^{1/2}U^\top$ and let $a_i$ be the $i$-th column of $A$, and $e_i$ be the $i$-th canonical basis vector for each $i \in N$. We prove that $\|a_i\|^2 = l_i(\lambda)$, the true leverage score, since

$$\|a_i\|^2 = \|Q_n^{1/2}U^\top e_i\|^2 = e_i^\top U Q_n U^\top e_i = ((K_n + \lambda n I)^{-1}K_n)_{ii} = l_i(\lambda).$$

Noting that $\sum_{k=1}^n \frac{q(k)}{P_\lambda(k)}a_k a_k^\top = \sum_{i=\Im} \frac{1}{P_\lambda(i)}a_i a_i^\top$, we have

$$\beta_3(\lambda) = \|AA^\top - \frac{1}{m}\sum_{i\in\Im}\frac{1}{P_\lambda(i)}a_i a_i^\top\|.$$

Moreover, by the $T$-approximation property of the approximate leverage scores (see Def. 1), we have that for all $i \in \{1, \ldots, n\}$, when $\lambda \geq \lambda_0$, the following holds with probability $1 - \delta$

$$P_\lambda(i) = \frac{\hat{l}_i(\lambda)}{\sum_j \hat{l}_j(\lambda)} \geq T^{-2}\frac{l_i(\lambda)}{\sum_j l_j(\lambda)} = T^{-2}\frac{\|a_i\|^2}{\operatorname{Tr} AA^\top}.$$

Then, we can apply Prop. 9, so that, after a union bound, we obtain the following inequality with probability $1 - \delta - \tau$:

$$\beta_3(\lambda) \leq \frac{2\|A\|^2 \log \frac{2n}{\tau}}{3m} + \sqrt{\frac{2\|A\|^2 T^2 \operatorname{Tr} AA^\top \log \frac{2n}{\tau}}{m}} \leq \frac{2\log \frac{2n}{\tau}}{3m} + \sqrt{\frac{2T^2 \hat{\mathcal{N}}(\lambda)\log \frac{2n}{\tau}}{m}},$$

where the last step follows from $\|A\|^2 = \|(K_n + \lambda n I)^{-1}K_n\| \leq 1$ and $\operatorname{Tr}(AA^\top) = \operatorname{Tr}(C_{n\lambda}^{-1}C_n) := \hat{\mathcal{N}}(\lambda)$. Applying Proposition 1, we have that $\hat{\mathcal{N}}(\lambda) \leq 1.3\mathcal{N}(\lambda)$ with probability $1 - \tau$, when $\frac{19\kappa^2}{n}\log\frac{n}{4\tau} \leq \lambda \leq \|C\|$ and $n \geq 405\kappa^2 \vee 67\kappa^2 \log\frac{\kappa^2}{2\tau}$. Thus, by taking a union bound again, we have

$$\beta_3(\lambda) \leq \frac{2\log\frac{2n}{\tau}}{3m} + \sqrt{\frac{5.3T^2\mathcal{N}(\lambda)\log\frac{2n}{\tau}}{m}},$$

with probability $1 - 2\tau - \delta$ when $\lambda_0 \vee \frac{19\kappa^2}{n} \log \frac{n}{\delta} \leq \lambda \leq \|C\|$ and $n \geq 405\kappa^2 \vee 67\kappa^2 \log \frac{2\kappa^2}{\delta}$. The last step is to bound $\|C_\lambda^{-1/2} C_{n\lambda}^{1/2}\|^2$, as follows

$$\|C_\lambda^{-1/2} C_{n\lambda}^{1/2}\|^2 = \|C_\lambda^{-1/2} C_{n\lambda} C_\lambda^{-1/2}\| = \|I + C_\lambda^{-1/2}(C_n - C)C_\lambda^{-1/2}\| \leq 1 + \eta,$$

with $\eta = \|C_\lambda^{-1/2}(C_n - C)C_\lambda^{-1/2}\|$. Note that, by applying Prop. 8 we have that $\eta \leq \frac{2(\kappa^2+\lambda)\theta}{3\lambda n} + \sqrt{\frac{2\kappa^2\theta}{3\lambda n}}$ with probability $1 - \tau$ and $\theta = \log \frac{8\kappa^2}{\lambda\tau}$. Finally, by collecting the above results and taking a union bound we have

$$\beta(\lambda) \leq \frac{2w}{3n} + \sqrt{\frac{2w\kappa^2}{\lambda n}} + (1 + \eta)\left(\frac{2\log\frac{2n}{\tau}}{3m} + \sqrt{\frac{5.3T^2\mathcal{N}(\lambda)\log\frac{2n}{\tau}}{m}}\right),$$

with probability $1 - 4\tau - \delta = 1 - 2\delta$ when $\lambda_0 \vee \frac{19\kappa^2}{n} \log \frac{n}{\delta} \leq \lambda \leq \|C\|$ and $n \geq 405\kappa^2 \vee 67\kappa^2 \log \frac{2\kappa^2}{\delta}$. Note that, if we select $n \geq 405\kappa^2 \vee 223\kappa^2 \log \frac{2\kappa^2}{\delta}$, $m \geq 334 \log \frac{8n}{\delta}$, $\lambda_0 \vee \frac{19\kappa^2}{n} \log \frac{2n}{\delta} \leq \lambda \leq \|C\|$ and $\frac{78T^2\mathcal{N}(\lambda)\log\frac{8n}{\delta}}{m} \leq 1$ the conditions are satisfied and we have $\beta(\lambda) \leq 2/3$, so that

$$\|(I - P_m)C^{1/2}\|^2 \leq 3\lambda,$$

with probability $1 - 2\delta$. $\qquad\square$

**Proposition 1** (Empirical Effective Dimension). *Let* $\hat{\mathcal{N}}(\lambda) = \operatorname{Tr} C_n C_{n\lambda}^{-1}$. *Under the Assumption 3, for any* $\delta > 0$ *and* $n \geq 405\kappa^2 \vee 67\kappa^2 \log \frac{6\kappa^2}{\delta}$, *if* $\frac{19\kappa^2}{n} \log \frac{n}{4\delta} \leq \lambda \leq \|C\|$, *then the following holds with probability* $1 - \delta$,

$$\frac{|\hat{\mathcal{N}}(\lambda) - \mathcal{N}(\lambda)|}{\mathcal{N}(\lambda)} \leq 4.5q + (1 + 9q)\sqrt{\frac{3q}{\mathcal{N}(\lambda)}} + \frac{q + 13.5q^2}{\mathcal{N}(\lambda)} \leq 1.65,$$

*with* $q = \frac{4\kappa^2 \log\frac{6}{\delta}}{3\lambda n}$.

*Proof.* Let $\tau = \delta/3$. Define $B_n = C_\lambda^{-1/2}(C - C_n)C_\lambda^{-1/2}$. Choosing $\lambda$ in the range $\frac{19\kappa^2}{n} \log \frac{n}{4\tau} \leq \lambda \leq \|C\|$, Prop. 8 assures that $\lambda_{\max}(B_n) \leq 1/3$ with probability $1 - \tau$. Then, using the fact that $C_{n\lambda}^{-1} = C_\lambda^{-1/2}(I - B_n)^{-1}C_\lambda^{-1/2}$ (see the proof of Prop. 7) we have

$$|\hat{\mathcal{N}}(\lambda) - \mathcal{N}(\lambda)| = |\operatorname{Tr} C_{n\lambda}^{-1}C_n - CC_\lambda^{-1} = \lambda \operatorname{Tr} C_{n\lambda}^{-1}(C_n - C)C_\lambda^{-1}|$$

$$= |\lambda \operatorname{Tr} C_\lambda^{-1/2}(I - B_n)^{-1} C_\lambda^{-1/2}(C_n - C)C_\lambda^{-1/2}C_\lambda^{-1/2}|$$

$$= |\lambda \operatorname{Tr} C_\lambda^{-1/2}(I - B_n)^{-1} B_n C_\lambda^{-1/2}|.$$

Considering that for any symmetric linear operator $X : \mathcal{H} \to \mathcal{H}$ the following identity holds

$$(I - X)^{-1}X = X + X(I - X)^{-1}X,$$

when $\lambda_{\max}(X) \leq 1$, we have

$$\lambda|\operatorname{Tr} C_\lambda^{-1/2}(I - B_n)^{-1} B_n C_\lambda^{-1/2}| \leq \underbrace{\lambda|\operatorname{Tr} C_\lambda^{-1/2} B_n C_\lambda^{-1/2}|}_{A}$$

$$+ \underbrace{\lambda|\operatorname{Tr} C_\lambda^{-1/2} B_n (I - B_n)^{-1} B_n C_\lambda^{-1/2}|}_{B}.$$

To find an upperbound for $A$ define the i.i.d. random variables $\eta_i = \langle K_{x_i}, \lambda C_\lambda^{-2} K_{x_i}\rangle \in \mathbb{R}$ with $i \in \{1, \ldots, n\}$. By linearity of the trace and the expectation, we have $M = \mathbb{E}\eta_1 = \mathbb{E}\langle K_{x_i}, \lambda C_\lambda^{-2} K_{x_i}\rangle = \mathbb{E}\operatorname{Tr}(\lambda C_\lambda^{-2} K_{x_1} \otimes K_{x_1}) = \lambda \operatorname{Tr}(C_\lambda^{-2}C)$. Therefore,

$$\lambda|\operatorname{Tr} C_\lambda^{-1/2} B_n C_\lambda^{-1/2}| = \left|M - \frac{1}{n}\sum_{i=1}^{n} \eta_i\right|,$$

and we can apply the Bernstein inequality (Prop. 10) with

$$|M - \eta_1| \leq \lambda \|C_\lambda^{-2}\| \|K_{x_1}\|^2 + M \leq \frac{\kappa^2}{\lambda} + M \leq \frac{2\kappa^2}{\lambda} = L,$$

$$\mathbb{E}(\eta_1 - M)^2 = \mathbb{E}\eta_1^2 - M^2 \leq \mathbb{E}\eta_1^2 \leq LM = \sigma^2.$$

An upperbound for $M$ is $M = \text{Tr}(\lambda C_\lambda^{-2}C) = \text{Tr}((I - C_\lambda^{-1}C)C_\lambda^{-1}C) \leq \mathcal{N}(\lambda)$. Thus, we have

$$\lambda |\text{Tr}\, C_\lambda^{-1/2}B_nC_\lambda^{-1/2}| \leq \frac{4\kappa^2 \log\frac{2}{\tau}}{3\lambda n} + \sqrt{\frac{4\kappa^2\mathcal{N}(\lambda)\log\frac{2}{\tau}}{\lambda n}},$$

with probability $1 - \tau$.

To find an upperbound for $B$, let $\mathcal{L}$ be the space of Hilbert-Schmidt operators on $\mathcal{H}$. $\mathcal{L}$ is a Hilbert space with scalar product $\langle U, V \rangle_{HS} = \text{Tr}\,(UV^*)$ for all $U, V \in \mathcal{L}$. Next, note that $B = \|Q\|_{HS}^2$ where $Q = \lambda^{1/2}C_\lambda^{-1/2}B_n\,(I - B_n)^{-1/2}$, moreover

$$\|Q\|_{HS}^2 \leq \|\lambda^{1/2}C_\lambda^{-1/2}\|^2 \|B_n\|_{HS}^2 \|(I - B_n)^{-1/2}\|^2 \leq 1.5\|B_n\|_{HS}^2,$$

since $\|(I - B_n)^{-1/2}\|^2 = (1 - \lambda_{\max}(B_n))^{-1} \leq 3/2$ and $(1 - \sigma)^{-1}$ is increasing and positive on $[-\infty, 1)$.

To find a bound for $\|B_n\|_{HS}$ consider that $B_n = T - \frac{1}{n}\sum_{i=1}^n \zeta_i$ where $\zeta_i$ are i.i.d. random operators defined as $\zeta_i = C_\lambda^{-1/2}(K_{x_i} \otimes K_{x_i})C_\lambda^{-1/2} \in \mathcal{L}$ for all $i \in \{1, \ldots, n\}$, and $T = \mathbb{E}\zeta_1 = C_\lambda^{-1}C \in \mathcal{L}$. Then we can apply the Bernstein's inequality for random vectors on a Hilbert space (Prop. 11), with the following $L$ and $\sigma^2$:

$$\|T - \zeta_1\|_{HS} \leq \|C_\lambda^{-1/2}\|^2 \|K_{x_1}\|_{\mathcal{H}}^2 + \|T\|_{HS} \leq \frac{\kappa^2}{\lambda} + \|T\|_{HS} \leq \frac{2\kappa^2}{\lambda} = L,$$

$$\mathbb{E}\|\zeta_1 - T\|^2 = \mathbb{E}\,\text{Tr}(\zeta_1^2 - T^2) \leq \mathbb{E}\,\text{Tr}(\zeta_1^2) \leq L\mathbb{E}\,\text{Tr}(\zeta_1) = \sigma^2,$$

where $\|T\|_{HS} \leq \mathbb{E}\,\text{Tr}(\zeta_1) = \mathcal{N}(\lambda)$, obtaining

$$\|B_n\|_{HS} \leq \frac{4\kappa^2 \log\frac{2}{\tau}}{\lambda n} + \sqrt{\frac{4\kappa^2\mathcal{N}(\lambda)\log\frac{2}{\tau}}{\lambda n}},$$

with probability $1 - \tau$. Then, by taking a union bound for the three events we have

$$|\hat{\mathcal{N}}(\lambda) - \mathcal{N}(\lambda)| \leq q + \sqrt{3q\mathcal{N}(\lambda)} + 1.5\left(3q + \sqrt{3q\mathcal{N}(\lambda)}\right)^2,$$

with $q = \frac{4\kappa^2 \log\frac{6}{\delta}}{3\lambda n}$, and with probability $1 - \delta$. Finally, if the second assumption on $\lambda$ holds, then we have $q \leq 4/57$. Noting that $n \geq 405\kappa^2$, and that $\mathcal{N}(\lambda) \geq \|CC_\lambda^{-1}\| = \frac{\|C\|}{\|C\|+\lambda} \geq 1/2$, we have that

$$|\hat{\mathcal{N}}(\lambda) - \mathcal{N}(\lambda)| \leq \left(\frac{q}{3\mathcal{N}(\lambda)} + \sqrt{\frac{q}{\mathcal{N}(\lambda)}} + 1.5\left(\frac{q}{\sqrt{\mathcal{N}(\lambda)}} + \sqrt{q}\right)^2\right)\mathcal{N}(\lambda) \leq 1.65\mathcal{N}(\lambda).$$

$\square$

# E   Proofs of main theorem

A key step to derive the proof of Theorem 1 is the error decomposition given by the following theorem, together with the probabilistic inequalities in the previous section.

**Theorem 2** (Error decomposition for KRLS+Ny). *Under Assumptions 1, 3, 4, let $v = \min(s, 1/2)$ and $\hat{f}_{\lambda,m}$ a KRLS + generalized Nyström solution as in Eq. (18). Then for any $\lambda, m > 0$ the error is bounded by*

$$\left|\mathcal{E}(\hat{f}_{\lambda,m}) - \mathcal{E}(f_{\mathcal{H}})\right|^{1/2} \leq q(\underbrace{\mathcal{S}(\lambda, n)}_{\textit{Sample error}} + \underbrace{\mathcal{C}(m)^{1/2+v}}_{\textit{Computational error}} + \underbrace{\lambda^{1/2+v}}_{\textit{Approximation error}}), \tag{21}$$

*where $\mathcal{S}(\lambda, n) = \|(C + \lambda I)^{-1/2}(S_n^*\hat{y}_n - C_nf_{\mathcal{H}})\|$ and $\mathcal{C}(m) = \|(I - P_m)(C + \lambda I)^{1/2}\|^2$ with $P_m = Z_m^*(Z_mZ_m^*)^\dagger Z_m$. Moreover $q = R(\beta^2 \vee (1 + \theta\beta))$, $\beta = \|(C_n + \lambda I)^{-1/2}(C + \lambda I)^{1/2}\|$, $\theta = \|(C_n + \lambda I)^{1/2}(C + \lambda I)^{-1/2}\|$.*

*Proof.* Let $C_\lambda = C + \lambda I$ and $C_{n\lambda} = C_n + \lambda I$ for any $\lambda > 0$. Let $\hat{f}_{\lambda,m}$ as in Eq. (18). By Lemma 1, Lemma 2 and Lemma 3 we know that $\hat{f}_{\lambda,m}$ is characterized by $\hat{f}_{\lambda,m} = g_{\lambda m}(C_n) S_n^* \hat{y}_n$ with $g_{\lambda,m}(C_n) = V(V^* C_n V + \lambda I)^{-1} V^*$. By using the fact that $\mathcal{E}(f) - \mathcal{E}(f_\mathcal{H}) = \|C^{1/2}(f - f_\mathcal{H})\|_\mathcal{H}^2$ for any $f \in \mathcal{H}$ (see Prop. 1 Point 3 of [7]), we have

$$
\begin{aligned}
|\mathcal{E}(\hat{f}_{\lambda,m}) - \mathcal{E}(f_\mathcal{H})|^{1/2} &= \|C^{1/2}(\hat{f}_{\lambda,m} - f_\mathcal{H})\|_\mathcal{H} = \|C^{1/2}(g_{\lambda,m}(C_n) S_n^* \hat{y}_n - f_\mathcal{H})\|_\mathcal{H} \\
&= \|C^{1/2}(g_{\lambda,m}(C_n) S_n^*(\hat{y}_n - S_n f_\mathcal{H} + S_n f_\mathcal{H}) - f_\mathcal{H})\|_\mathcal{H} \\
&\leq \underbrace{\|C^{1/2} g_{\lambda,m}(C_n) S_n^*(\hat{y}_n - S_n f_\mathcal{H})\|_\mathcal{H}}_{A} + \underbrace{\|C^{1/2}(I - g_{\lambda,m}(C_n) C_n) f_\mathcal{H}\|_\mathcal{H}}_{B}.
\end{aligned}
$$

**Bound for the term A** Multiplying and dividing by $C_{n\lambda}^{1/2}$ and $C_\lambda^{1/2}$ we have

$$
A \leq \|C^{1/2} C_{n\lambda}^{-1/2}\|\|C_{n\lambda}^{1/2} g_{\lambda,m}(C_n) C_{n\lambda}^{1/2}\|\|C_{n\lambda}^{-1/2} C_\lambda^{1/2}\|\|C_\lambda^{-1/2} S_n^*(\hat{y}_n - S_n f_\mathcal{H})\|_\mathcal{H} \leq \beta^2 \, \mathcal{S}(\lambda, n),
$$

where the last step is due to Lemma 8 and the fact that

$$
\|C^{1/2} C_{n\lambda}^{-1/2}\| \leq \|C^{1/2} C_\lambda^{-1/2}\|\|C_\lambda^{1/2} C_{n\lambda}^{-1/2}\| \leq \|C_\lambda^{1/2} C_{n\lambda}^{-1/2}\|.
$$

**Bound for the term B** Noting that $g_{\lambda,m}(C_n) C_{n\lambda} V V^* = V V^*$, we have

$$
\begin{aligned}
I - g_{\lambda,m}(C_n) C_n &= I - g_{\lambda,m}(C_n) C_{n\lambda} + \lambda g_{\lambda,m}(C_n) \\
&= I - g_{\lambda,m}(C_n) C_{n\lambda} V V^* - g_{\lambda,m}(C_n) C_{n\lambda}(I - V V^*) + \lambda g_{\lambda,m}(C_n) \\
&= (I - V V^*) + \lambda g_{\lambda,m}(C_n) - g_{\lambda,m}(C_n) C_{n\lambda}(I - V V^*).
\end{aligned}
$$

Therefore, noting that by Ass. 4 we have $\|C_\lambda^{-v} f_\mathcal{H}\|_\mathcal{H} \leq \|C_\lambda^{-s} f_\mathcal{H}\|_\mathcal{H} \leq \|C^{-s} f_\mathcal{H}\|_\mathcal{H} \leq R$, then, by reasoning as in A, we have

$$
\begin{aligned}
B &\leq \|C^{1/2}(I - g_{\lambda,m}(C_n) C_n) C_\lambda^v\|\|C_\lambda^{-v} f_\mathcal{H}\|_\mathcal{H} \\
&\leq R\|C^{1/2} C_\lambda^{-1/2}\|\|C_\lambda^{1/2}(I - V V^*) C_\lambda^v\| + R\lambda\|C^{1/2} C_{n\lambda}^{-1/2}\|\|C_{n\lambda}^{1/2} g_{\lambda,m}(C_n) C_\lambda^v\| \\
&\quad + R\|C^{1/2} C_{n\lambda}^{-1/2}\|\|C_{n\lambda}^{1/2} g_{\lambda,m}(C_n) C_{n\lambda}^{1/2}\|\|C_{n\lambda}^{1/2} C_\lambda^{-1/2}\|\|C_\lambda^{1/2}(I - V V^*) C_\lambda^v\| \\
&\leq R(1 + \beta\theta) \underbrace{\|C_\lambda^{1/2}(I - V V^*) C_\lambda^v\|}_{B.1} + R\beta \underbrace{\lambda\|C_{n\lambda}^{1/2} g_{\lambda,m}(C_n) C_\lambda^v\|}_{B.2},
\end{aligned}
$$

where in the second step we applied the decomposition of $I - g_{\lambda m}(C_n) C_n$.

**Bound for the term B.1** Since $V V^*$ is a projection operator, we have that $(I - V V^*) = (I - V V^*)^s$, for any $s > 0$, therefore

$$
B.1 = \|C_\lambda^{1/2}(I - V V^*)^2 C_\lambda^v\| \leq \|C_\lambda^{1/2}(I - V V^*)\|\|(I - V V^*) C_\lambda^v\|.
$$

By applying Cordes inequality (Prop. 4) to $\|(I - V V^*) C_\lambda^v\|$ we have,

$$
\|(I - V V^*) C_\lambda^v\| = \|(I - V V^*)^{2v} C_\lambda^{\frac{1}{2} 2v}\| = \|(I - V V^*) C_\lambda^{1/2}\|^{2v}.
$$

**Bound for the term B.2** We have

$$
\begin{aligned}
B.2 &\leq \lambda\|C_{n\lambda}^{1/2} g_{\lambda,m}(C_n) C_{n\lambda}^v\|\|C_{n\lambda}^{-v} C_\lambda^v\| \\
&\leq \lambda\|C_{n\lambda}^{1/2} g_{\lambda,m}(C_n) C_{n\lambda}^v\|\|C_{n\lambda}^{-1/2} C_\lambda^{1/2}\|^{2v} \\
&\leq \beta^{2v} \lambda\|(V^* C_{n\lambda} V)^{1/2}(V^* C_{n\lambda} V)^{-1}(V^* C_{n\lambda} V)^v\| \\
&= \beta^{2v} \lambda\|(V^* C_n V + \lambda I)^{-(1/2-v)}\| \leq \beta\lambda^{1/2+v},
\end{aligned}
$$

where the first step is obtained multipling and dividing by $C_{n\lambda}^v$, the second step by applying Cordes inequality (see Prop. 4), the third step by Prop. 6. $\qquad\square$

**Proposition 2** (Bounds for plain and ALS Nyström). *For any $\delta > 0$, let $n \geq 1655\kappa^2 + 223\kappa^2 \log \frac{6\kappa^2}{\delta}$, let $\frac{19\kappa^2}{n} \log \frac{6n}{\delta} \leq \lambda \leq \|C\|$ and define*

$$
\mathcal{C}_{\mathrm{pl}}(m) = \min\left\{ t > 0 \,\middle|\, (67 \vee 5\mathcal{N}_\infty(t)) \log \frac{12\kappa^2}{t\delta} \leq m \right\},
$$

$$
\mathcal{C}_{\mathrm{ALS}}(m) = \min\left\{ \frac{19\kappa^2}{n} \log \frac{12n}{\delta} \leq t \leq \|C\| \,\middle|\, 78T^2\mathcal{N}(t) \log \frac{48n}{\delta} \leq m \right\}.
$$

*Under the assumptions of Thm. 2 and Assumption 2, 3, if one of the following two conditions hold*

1. *plain Nyström is used,*

2. *ALS Nyström is used with*

    (a) *$T$-approximate leverage scores, for any $t \geq \frac{19\kappa^2}{n} \log \frac{12n}{\delta}$ (see Def. 1),*

    (b) *resampling probabilities $P_t$ where $t = \mathcal{C}_{ALS}(m)$ (see Sect. 2),*

    (c) *$m \geq 334 \log \frac{48n}{\delta}$,*

*then the following holds with probability $1 - \delta$*

$$\left| \mathcal{E}(\hat{f}_{\lambda,m}) - \mathcal{E}(f_{\mathcal{H}}) \right|^{1/2} \leq 6R \left( \frac{M\sqrt{\mathcal{N}_\infty(\lambda)}}{n} + \sqrt{\frac{\sigma^2 \mathcal{N}(\lambda)}{n}} \right) \log \frac{6}{\delta} + 3R\mathcal{C}(m)^{1/2+v} + 3R\lambda^{1/2+v}$$

$$(22)$$

*where $\mathcal{C}(m) = \mathcal{C}_{\mathrm{pl}}(m)$ in case of plain Nyström and $\mathcal{C}(m) = \mathcal{C}_{\mathrm{ALS}}(m)$ in case of ALS Nyström.*

*Proof.* In order to get explicit bounds from Thm. 2, we have to control four quantities that are $\beta, \theta, \mathcal{S}(\lambda, n)$ and $\mathcal{C}(m)$. In the following we bound such quantities in probability and then take a union bound. Let $\tau = \delta/3$. We can control both $\beta$ and $\theta$, by bounding $b(\lambda) = \|C_\lambda^{-1/2}(C_n - C)C_\lambda^{-1/2}\|$. Indeed, by Prop. 7, we have that $\beta \leq 1/(1 - b(\lambda))$, while

$$\theta^2 = \|C_\lambda^{-1/2} C_{n\lambda} C_\lambda^{-1/2}\| = \|I + C_\lambda^{-1/2}(C_n - C)C_\lambda^{-1/2}\| \leq 1 + b(\lambda).$$

Exploiting Prop. 8, with the fact that $\mathcal{N}(\lambda) \leq \mathcal{N}_\infty(\lambda) \leq \frac{\kappa^2}{\lambda}$ and $\mathrm{Tr}\, C \leq \kappa^2$, we have that $b(\lambda) \leq \frac{2(\kappa^2 + \lambda)w}{3\lambda n} + \sqrt{\frac{2w\kappa^2}{\lambda n}}$ for $w = \log \frac{4\kappa^2}{\tau\lambda}$ with probability $1 - \tau$. Simple computations show that with $n$ and $\lambda$ as in the statement of this corollary, we have $b(\lambda) \leq 1/3$. Therefore $\beta \leq 1.5$, while $\theta \leq 1.16$ and $q = R(\beta^2 \vee (1 + \theta\beta)) < 2.75R$ with probability $1 - \tau$. Next, we bound $\mathcal{S}(\lambda, n)$. Here we exploit Lemma 4 which gives, with probability $1 - \tau$,

$$\mathcal{S}(\lambda, n) \leq 2 \left( \frac{M\sqrt{\mathcal{N}_\infty(\lambda)}}{n} + \sqrt{\frac{\sigma^2 \mathcal{N}(\lambda)}{n}} \right) \log \frac{2}{\tau}.$$

To bound $\mathcal{C}(m)$ for plain Nyström, Lemma 6 gives $\mathcal{C}(m) \leq 3t$ with probability $1 - \tau$, for a $t > 0$ such that $(67 \vee 5\mathcal{N}_\infty(t)) \log \frac{4\kappa^2}{t\tau} \leq m$. In particular, we choose $t = \mathcal{C}_{\mathrm{pl}}(m)$ to satisfy the condition. Next we bound $\mathcal{C}(m)$ for ALS Nyström. Using Lemma 7 with $\lambda_0 = \frac{19\kappa^2}{n} \log \frac{2n}{\tau}$, we have $\mathcal{C}(m) \leq 3t$ with probability $1 - \tau$ under some conditions on $t, m, n$, on the approximate leverage scores and on the resampling probability. Here again the requirement on $n$ is satisfied by the hypotesis on $n$ of this proposition, while the condition on the approximate leverage scores and on the resampling probabilities are satisfied by conditions (a), (b) of this proposition. The remaining two conditions are $\frac{19\kappa^2}{n} \log \frac{4n}{\tau} \leq t \leq \|C\|$ and $(334 \vee 78T^2\mathcal{N}(t)) \log \frac{16n}{\tau} \leq m$. They are satisfied by choosing $t = \mathcal{C}_{\mathrm{ALS}}(m)$ and by assuming that $m \geq 334 \log \frac{16n}{\tau}$. Finally, the proposition is obtained by substituting each of the four quantities $\beta, \theta, \mathcal{S}(\lambda, n), \mathcal{C}(m)$ with the corresponding upperbounds in Eq. (21), and by taking the union bounds on the associated events. □

*Proof of Theorem 1.* By exploiting the results of Prop. 2, obtained from the error decomposition of Thm. 2 we have that

$$\left| \mathcal{E}(\hat{f}_{\lambda,m}) - \mathcal{E}(f_{\mathcal{H}}) \right|^{1/2} \leq 6R \left( \frac{M\sqrt{\mathcal{N}_\infty(\lambda)}}{n} + \sqrt{\frac{\sigma^2 \mathcal{N}(\lambda)}{n}} \right) \log \frac{6}{\delta} + 3R\mathcal{C}(m)^{1/2+v} + 3R\lambda^{1/2+v}$$

$$(23)$$

with probability $1 - \delta$, under conditions on $\lambda, m, n$, on the resampling probabilities and on the approximate leverage scores. The last is satisfied by condition (a) in this theorem. The conditions on $\lambda, n$ are $n \geq 1655\kappa^2 + 223\kappa^2 \log \frac{6\kappa^2}{\delta}$ and $\frac{19\kappa^2}{n} \log \frac{12n}{\delta} \leq \lambda \leq \|C\|$. If we assume that $n \geq 1655\kappa^2 + 223\kappa^2 \log \frac{6\kappa^2}{\delta} + \left( \frac{38p}{\|C\|} \log \frac{114\kappa^2 p}{\|C\|\delta} \right)^p$ we satisfy the condition on $n$ and at the same time we are sure that $\lambda = \|C\| n^{-1/(2v+\gamma+1)}$ satisfies the condition on $\lambda$. In the plain Nyström case,

if we assume that $m \geq 67 \log \frac{12\kappa^2}{\lambda\delta} + 5\mathcal{N}_\infty(\lambda) \log \frac{12\kappa^2}{\lambda\delta}$, then $\mathcal{C}(m) = \mathcal{C}_{\mathrm{pl}}(m) \leq \lambda$. In the ALS Nyström case, if we assume that $m \geq (334 \vee 78T^2\mathcal{N}(\lambda)) \log \frac{48n}{\delta}$ the condition on $m$ is satisfied, then $\mathcal{C}(m) = \mathcal{C}_{\mathrm{ALS}}(m) \leq \lambda$, moreover the conditions on the resampling probabilities is satisfied by condition (b) of this theorem. Therefore, by setting $\lambda = \|C\| n^{-1/(2v+\gamma+1)}$ in Eq. (23) and considering that $\mathcal{N}_\infty(\lambda) \leq \kappa^2\lambda^{-1}$ we easily obtain the result of this theorem. $\qquad\square$

The following lemma is a technical result needed in the error decomposition (Thm. 2).

**Lemma 8.** *For any $\lambda > 0$, let $V$ be such that $V^*V = I$ and $C_n$ be a positive self-adjoint operator. Then, the following holds,*

$$\|(C_n + \lambda I)^{1/2} V (V^*C_n V + \lambda I)^{-1} V^* (C_n + \lambda I)^{1/2}\| \leq 1.$$

*Proof.* Let $C_{n\lambda} = C_n + \lambda I$ and $g_{\lambda m}(C_n) = V(V^*C_n V + \lambda I)^{-1}V^*$, then

$$\begin{aligned}
\|C_{n\lambda}^{1/2} g_{\lambda m}(C_n) C_{n\lambda}^{1/2}\|^2 &= \|C_{n\lambda}^{1/2} g_{\lambda m}(C_n) C_{n\lambda} g_{\lambda m}(C_n) C_{n\lambda}^{1/2}\| \\
&= \|C_{n\lambda}^{1/2} V (V^*C_{n\lambda}V)^{-1} (V^*C_{n\lambda}V)(V^*C_{n\lambda}V)^{-1}V^* C_{n\lambda}^{1/2}\| \\
&= \|C_{n\lambda}^{1/2} g_{\lambda m}(C_n) C_{n\lambda}^{1/2}\|,
\end{aligned}$$

and therefore the only possible values for $\|C_{n\lambda}^{1/2} g_{\lambda m}(C_n) C_{n\lambda}^{1/2}\|$ are 0 or 1. $\qquad\square$

## F  Auxiliary results

**Proposition 3.** *Let $\mathcal{H}, \mathcal{K}, \mathcal{F}$ three separable Hilbert spaces, let $Z : \mathcal{H} \to \mathcal{K}$ be a bounded linear operator and let $W$ be a projection operator on $\mathcal{H}$ such that $\mathrm{ran}\, P = \overline{\mathrm{ran}\, Z^*}$. Then for any bounded linear operator $F : \mathcal{F} \to \mathcal{H}$ and any $\lambda > 0$ we have*

$$\|(I - P)X\| \leq \lambda^{1/2}\|(Z^*Z + \lambda I)^{-1/2}X\|.$$

*Proof.* First of all note that $\lambda(Z^*Z + \lambda I)^{-1} = I - Z^*(ZZ^* + \lambda I)^{-1}Z$, that $Z = ZP$ and that $\|Z^*(ZZ^* + \lambda I)^{-1}Z\| \leq 1$ for any $\lambda > 0$. Then for any $v \in \mathcal{H}$ we have

$$\begin{aligned}
\langle v, Z^*(ZZ^* + \lambda I)^{-1}Zv \rangle &= \langle v, PZ^*(ZZ^* + \lambda I)^{-1}ZPv \rangle = \|(ZZ^* + \lambda I)^{-1/2}ZPv\|^2 \\
&\leq \|(ZZ^* + \lambda I)^{-1/2}Z\|^2 \|Pv\|^2 \leq \|Pv\|^2 = \langle v, Pv \rangle
\end{aligned}$$

therefore $P - Z^*(ZZ^* + \lambda I)^{-1}Z$ is a positive operator, and $(I - Z^*(ZZ^* + \lambda I)^{-1}Z) - (I - P)$ too. Now we can apply Prop. 5. $\qquad\square$

**Proposition 4** (Cordes Inequality [9])**.** *Let $A, B$ two positive semidefinite bounded linear operators on a separable Hilbert space. Then*

$$\|A^s B^s\| \leq \|AB\|^s \quad \text{when } 0 \leq s \leq 1.$$

**Proposition 5.** *Let $\mathcal{H}, \mathcal{K}, \mathcal{F}, \mathcal{G}$ be three separable Hilbert spaces and let $X : \mathcal{H} \to \mathcal{K}$ and $Y : \mathcal{H} \to \mathcal{F}$ be two bounded linear operators. For any bounded linear operator $Z : \mathcal{G} \to \mathcal{H}$, if $Y^*Y - X^*X$ is a positive self-adjoint operator then $\|XZ\| \leq \|YZ\|$.*

*Proof.* If $Y^*Y - X^*X$ is a positive operator then $Z^*(Y^*Y - X^*X)Z$ is positive too. Thus for all $f \in \mathcal{H}$ we have that $\langle f, (Q - P)f \rangle \geq 0$, where $Q = Z^*Y^*YZ$ and $P = Z^*X^*XZ$. Thus, by linearity of the inner product, we have

$$\|Q\| = \sup_{f \in \mathcal{G}} \langle f, Qf \rangle = \sup_{f \in \mathcal{G}} \{\langle f, Pf \rangle + \langle f, (Q - P)f \rangle\} \geq \sup_{f \in \mathcal{G}} \langle f, Pf \rangle = \|P\|.$$

$\qquad\square$

**Proposition 6.** *Let $\mathcal{H}, \mathcal{K}$ be two separable Hilbert spaces, let $A : \mathcal{H} \to \mathcal{H}$ be a positive linear operator, $V : \mathcal{H} \to \mathcal{K}$ a partial isometry and $B : \mathcal{K} \to \mathcal{K}$ a bounded operator. Then $\|A^r V B V^* A^s\| \leq \|(V^*AV)^r B (V^*AV)^s\|$, for all $0 \leq r, s \leq 1/2$.*

*Proof.* By Hansen's inequality (see [10]) we know that $(V^*AV)^{2t} - V^*A^{2t}V$ is positive selfadjoint operator for any $0 \le t \le 1/2$, therefore we can apply Prop. 5 two times, obtaining

$$\|A^r V (B V^* A^s)\| \le \|(V^*AV)^r (B V^* A^s)\| = \|((V^*AV)^r B) V^* A^s\| \le \|((V^*AV)^r B)(V^*AV)^s\|.$$

$\square$

**Proposition 7.** *Let $\mathcal{H}$ be a separable Hilbert space, let $A$, $B$ two bounded self-adjoint positive linear operators and $\lambda > 0$. Then*

$$\|(A + \lambda I)^{-1/2} B^{1/2}\| \le (1 - \beta)^{-1/2}$$

*when*

$$\beta = \lambda_{\max}\left[(B + \lambda I)^{-1/2}(B - A)(B + \lambda I)^{-1/2}\right] < 1.$$

*Proof.* Let $B_\lambda = B + \lambda I$. Note that

$$
\begin{aligned}
(A + \lambda I)^{-1} &= [(B + \lambda I) - (B - A)]^{-1} \\
&= \left[B_\lambda^{1/2}\left(I - B_\lambda^{-1/2}(B - A)B_\lambda^{-1/2}\right)B_\lambda^{1/2}\right]^{-1} \\
&= B_\lambda^{-1/2}\left[I - B_\lambda^{-1/2}(B - A)B_\lambda^{-1/2}\right]^{-1}B_\lambda^{-1/2}.
\end{aligned}
$$

Now let $X = (I - B_\lambda^{-1/2}(B - A)B_\lambda^{-1/2})^{-1}$. We have that,

$$
\begin{aligned}
\|(A + \lambda I)^{-1/2}B^{1/2}\| &= \|B^{1/2}(A + \lambda I)^{-1}B^{1/2}\|^{1/2} \\
&= \|B^{1/2}B_\lambda^{-1/2}XB_\lambda^{-1/2}B^{1/2}\|^{1/2} \\
&= \|X^{1/2}B_\lambda^{-1/2}B^{1/2}\|,
\end{aligned}
$$

because $\|Z\| = \|Z^*Z\|^{1/2}$ for any bounded operator $Z$. Note that

$$\|X^{1/2}B_\lambda^{-1/2}B^{1/2}\| \le \|X\|^{1/2}\|B_\lambda^{-1/2}B^{1/2}\| \le \|X\|^{1/2}.$$

Finally let $Y = B_\lambda^{-1/2}(B - A)B_\lambda^{-1/2}$ and assume that $\lambda_{\max}(Y) < 1$, then

$$\|X\| = \|(I - Y)^{-1}\| = (1 - \lambda_{\max}(Y))^{-1},$$

since $X = w(Y)$ with $w(\sigma) = (1 - \sigma)^{-1}$ for $-\infty \le \sigma < 1$, and $w$ is positive and monotonically increasing on the domain. $\square$

## G   Tail bounds

Let $\|\cdot\|_{HS}$ denote the Hilbert-Schmidt norm.

**Proposition 8.** *Let $v_1, \ldots, v_n$ with $n \ge 1$, be independent and identically distributed random vectors on a separable Hilbert spaces $\mathcal{H}$ such that $Q = \mathbb{E}\, v \otimes v$ exists, is trace class, and for any $\lambda > 0$ there exists a constant $\mathcal{N}_\infty(\lambda) < \infty$ such that $\langle v, (Q + \lambda I)^{-1}v\rangle \le \mathcal{N}_\infty(\lambda)$ almost everywhere. Let $Q_n = \frac{1}{n}\sum_{i=1}^n v_i \otimes v_i$ and take $0 < \lambda \le \|Q\|$. Then for any $\delta \ge 0$, the following holds*

$$\|(Q + \lambda I)^{-1/2}(Q - Q_n)(Q + \lambda I)^{-1/2}\| \le \frac{2\beta(1 + \mathcal{N}_\infty(\lambda))}{3n} + \sqrt{\frac{2\beta\mathcal{N}_\infty(\lambda)}{n}}$$

*with probability $1 - 2\delta$. Here $\beta = \log \frac{4\,\mathrm{Tr}\,Q}{\lambda\delta}$. Moreover it holds that*

$$\lambda_{\max}\left((Q + \lambda I)^{-1/2}(Q - Q_n)(Q + \lambda I)^{-1/2}\right) \le \frac{2\beta}{3n} + \sqrt{\frac{2\beta\mathcal{N}_\infty(\lambda)}{n}}$$

*with probability $1 - \delta$.*

*Proof.* Let $Q_\lambda = Q + \lambda I$. Here we apply Prop. 12 on the random variables $Z_i = M - Q_\lambda^{-1/2} v_i \otimes Q_\lambda^{-1/2} v_i$ with $M = Q_\lambda^{-1/2} Q Q_\lambda^{-1/2}$ for $1 \leq i \leq n$. Note that the expectation of $Z_i$ is 0. The random vectors are bounded by

$$\|Q_\lambda^{-1/2} Q Q_\lambda^{-1/2} - Q_\lambda^{-1/2} v_i \otimes Q_\lambda^{-1/2} v_i\| \leq \langle v, Q_\lambda^{-1} v \rangle + \|Q_\lambda^{-1/2} Q Q_\lambda^{-1/2}\| \leq \mathcal{N}_\infty(\lambda) + 1$$

and the second orded moment is

$$\mathbb{E}(Z_1)^2 = \mathbb{E} \ \langle v_1, Q_\lambda^{-1} v_1 \rangle \ Q_\lambda^{-1/2} v_1 \otimes Q_\lambda^{-1/2} v_1 \quad - \quad Q_\lambda^{-2} Q^2$$

$$\leq \mathcal{N}_\infty(\lambda) \mathbb{E} Q_\lambda^{-1/2} v_1 \otimes Q_\lambda^{-1/2} v_1 = \mathcal{N}_\infty(\lambda) Q = S.$$

Now we can apply Prop. 12. Now some considerations on $\beta$. It is $\beta = \log \frac{4 \operatorname{Tr} S}{\|S\|\delta} = \frac{4 \operatorname{Tr} Q_\lambda^{-1} Q}{\|Q_\lambda^{-1} Q\|\delta}$, now $\operatorname{Tr} Q_\lambda^{-1} Q \leq \frac{1}{\lambda} \operatorname{Tr} Q$. We need a lowerbound for $\|Q_\lambda^{-1} Q\| = \frac{\sigma_1}{\sigma_1 + \lambda}$ where $\sigma_1 = \|Q\|$ is the biggest eigenvalue of $Q$, now $\lambda \leq \sigma_1$ thus $\frac{\operatorname{Tr} Q}{\lambda \delta}$.

For the second bound of this proposition, the analysis remains the same except for $L$, indeed

$$\sup_{f \in \mathcal{H}} \langle f, Z_1 f \rangle = \sup_{f \in \mathcal{H}} \langle f, Q_\lambda^{-1} Q f \rangle - \left\langle f, Q_\lambda^{-1/2} v_i \right\rangle^2 \leq \sup_{f \in \mathcal{H}} \langle f, Q_\lambda^{-1} Q f \rangle \leq 1.$$

$\square$

**Remark 1.** *In Prop. 8, let define $\kappa^2 = \inf_{\lambda > 0} \mathcal{N}_\infty(\lambda)(\|Q\| + \lambda)$. When $n \geq 405\kappa^2 \vee 67\kappa^2 \log \frac{\kappa^2}{2\delta}$ and $\frac{9\kappa^2}{n} \log \frac{n}{2\delta} \leq \lambda \leq \|Q\|$ we have that*

$$\lambda_{\max} \left( (Q + \lambda I)^{-1/2} (Q - Q_n)(Q + \lambda I)^{-1/2} \right) \leq \frac{1}{2},$$

*with probability $1 - \delta$, while it is less than $1/3$ with the same probability, if $\frac{19\kappa^2}{n} \log \frac{n}{4\delta} \leq \lambda \leq \|Q\|$.*

**Proposition 9** (Theorem 2 [11]. Approximation of matrix products.). *Let $n, n$ be positive integers. Consider a matrix $A \in \mathbb{R}^{n \times n}$ and denote by $a_i$ the $i$-th column of $A$. Let $m \leq n$ and $I = \{i_1, \ldots, i_m\}$ be a subset of $N = \{1, \ldots, n\}$ formed by $m$ elements chosen randomly with replacement, according to a distribution that associates the probability $P(i)$ to the element $i \in N$. Assume that there exists a $\beta \in (0,1]$ such that the probabilities $P(1), \ldots, P(n)$ satisfy $P(i) \geq \beta \frac{\|a_i\|^2}{\operatorname{Tr} AA^\top}$ for all $i \in N$. For any $\delta > 0$ the following holds*

$$\|AA^\top - \frac{1}{m} \sum_{i \in I} \frac{1}{P(i)} a_i a_i^\top\| \leq \frac{2L \log \frac{2n}{\delta}}{3m} + \sqrt{\frac{2LS \log \frac{2n}{\delta}}{m}}$$

*with probability $1 - \delta$. Here $L = \|A\|^2$ and $S = \frac{1}{\beta} \operatorname{Tr} AA^\top$.*

**Proposition 10** (Bernstein's inequality for sum of random variables). *Let $x_1, \ldots, x_n$ be a sequence of independent and identically distributed random variables on $\mathbb{R}$ with zero mean. If there exists an $L, S \in \mathbb{R}$ such that $x_1 \leq L$ almost everywhere and $\mathbb{E} x_1^2 \leq S$, then for any $\delta > 0$ the following holds with probability $1 - \delta$:*

$$\frac{1}{n} \sum_{i=1}^n x_i \leq \frac{2L \log \frac{1}{\delta}}{3n} + \sqrt{\frac{2S \log \frac{1}{\delta}}{n}}.$$

*If there exists an $L' \geq |x_1|$ almost everywhere, then the same bound, computed with $L'$ instead of $L$, holds for the for the absolute value of the left hand side, with probability $1 - 2\delta$.*

*Proof.* It is a restatement of Theorem 3 of [12]. $\square$

**Proposition 11** (Bernstein's inequality for sum of random vectors). *Let $z_1, \ldots, z_n$ be a sequence of independent identically distributed random vectors on a separable Hilbert space $\mathcal{H}$. Assume $\mu = \mathbb{E} z_1$ exists and let $\sigma, M \geq 0$ such that*

$$\mathbb{E} \|z_1 - \mu\|_{\mathcal{H}}^p \leq \frac{1}{2} p! \sigma^2 L^{p-2}$$

*for all $p \geq 2$. Then for any $\tau \geq 0$:*

$$\|\frac{1}{n}\sum_{i=1}^{n} z_i - \mu\|_{\mathcal{H}} \leq \frac{2L\log\frac{2}{\delta}}{n} + \sqrt{\frac{2\sigma^2\log\frac{2}{\delta}}{n}}$$

*with probability greater or equal $1 - \delta$.*

*Proof.* restatement of Theorem 3.3.4 of [13]. □

**Proposition 12** (Bernstein's inequality for sum of random operators). *Let $\mathcal{H}$ be a separable Hilbert space and let $X_1, \ldots, X_n$ be a sequence of independent and identically distributed self-adjoint positive random operators on $\mathcal{H}$. Assume that there exists $\mathbb{E}X_1 = 0$ and $\lambda_{\max}(X_1) \leq L$ almost everywhere for some $L > 0$. Let $S$ be a positive operator such that $\mathbb{E}(X_1)^2 \leq S$. Then for any $\delta \geq 0$ the following holds*

$$\lambda_{\max}\left(\frac{1}{n}\sum_{i=1}^{n} X_i\right) \leq \frac{2L\beta}{3n} + \sqrt{\frac{2\|S\|\beta}{n}}$$

*with probability $1 - \delta$. Here $\beta = \log\frac{2\operatorname{Tr}S}{\|S\|\delta}$.*

*If there exists an $L'$ such that $L' \geq \|X_1\|$ almost everywhere, then the same bound, computed with $L'$ instead of $L$, holds for the operatorial norm with probability $1 - 2\delta$.*

*Proof.* The theorem is a restatement of Theorem 7.3.1 of [14] generalized to the separable Hilbert space case by means of the technique in Section 4 of [15]. □