[Reviews · NeurIPS 2015]

Submitted by Assigned_Reviewer_1

Comments: ---------

The paper investigaes statistical properties of two variants of the

Nystroem method for approximately solving the optimization problem of kernel least squares regression (KLSR), aka "least squares svm". The main result in this respect is Theorem 1 that establishes

optimal leaning rates under the "usual" Assumptions 1 - 4. Based on the way these rates are proved, the role of the subsample size as an additional regularization is discussed and, on small data sets, experimentally illustrated. Some experiments on large, standard datasets are reported, too.

The paper is well written and the exposition is, modulo some

small issues (see below) very clear. Although the proofs rely

on some techniques that have been previously used (e.g. for

investigating plain KLSR), the main result is extremely

interesting, significant, and novel.

Minor Remarks: --------------

- Assumption 1.

As far as I remember it has been claimed in several papers that

this assumption is superfluous, but I cannot remember any paper,

where this has actually been done. Please correct me if I am

wrong by adding a corresponding reference or explain in a bit more

detail how the assumption can be removed.

- Discussion following Assumption 4. I think it would be great for the reader if you can translate this assumption into an equivalent assumption on the image a fractional integral operator, since this is more standard. In addition, I think the reference [25] should be replaced by

S. Smale and D.X. Zhou, D.-X. (2003)

Estimating the approximation error in learning theory

I.Steinwart and C. Scovel (2012) Mercer's theorem on general domains: on the interaction between

measures, kernels, and RKHSs.

- Disccusion following Thereom 1. You should explain in more detail how your rates can be compared to

the lower bounds of [26,28]. In particular the comparison with your and their assumptions should be more detailed.

- Beginning Section 4.2 I did not understand which fraction you used for training, validation

and testing. Moreover, what exactly was repeated several times (e.g. was the test set always the same or not)? In addition, how did you

determine the values for sigma in your fest set of experiments?

Please also mention that while m can be somewhat small, its necessary

size is unknown in front, so that some sort of (costly) parameter selection seems to be necessary. Finally, you should also report time measurements on your large scale experiments (for both the parameter selection phase and the final

training phase). There seems to be plenty of space left in Figure 2 to do this.
Summary: The paper establishes optimal learning rates for two variants of

the Nystroem method for kernel least squares regression. It further discusses the regularization role of the subsample size and reports some experiments.

Submitted by Assigned_Reviewer_2

Nystrom-based approaches for kernel methods consists in subsampling columns/rows of the Gram matrix to cope with the computational and memory demands of kernel methods. So far, they have essentially been used for their computational benefits. The present paper proposes to study the regularization effect of the Nystrom approach.

The provided results concern Kernel Regularized Least Squares combined with Nystrom approximations. The main theorem shows how the computational gains of the Nystrom approximation appears in the generalization bound of KLRS. In addition, an algorithmic trick based on matrix block inversion, show how the "regularization path" i.e. a series of subsampling levels, can be computed efficiently.

As far as I have checked, the maths are correct. In addition, the authors have taken time to comment on the various assumptions provided as well on the theorems provided. It is a nice idea to have provided a peak on the proof of the main results.

Having the theoretical results supported by numerical simulations is also a good point of the paper.

All in all, this is an original paper, very well-written, with a significant contribution from a technical perspective.

Typos: - l.252: THE the proposed methodS is - l.280: approah - l.319: Eq.5 -> Eq. 5
Summary: The paper provides a rigorous analysis of the blessings, from the generalization viewpoint, of Nystrom-based sampling methods used to reduce the computational cost of kernel methods. The theoretical findings are supported by numerical simulations.

Submitted by Assigned_Reviewer_3

I very much enjoyed reading this paper.

The authors develop a non-trivial generalisation of previous analyses and give bounds that are optimal in the minimax sense. Moreover the bounds are meaningful and show that Nystrom approximation not only serves as a computational speedup technique but also has an implicit effect of regularisation, interacting with the L2 regularisation parameter.

The implicit regularisation effect of randomised matrix approximations in general was informally observed previously in [Mahoney: Randomized algorithms for matrices and data, 2011], but no precise explanation could be given. For random projection based approximation there are some attempts to quantify this effect to some extent in special cases e.g. in [Kaban: AISTATS2014] but for the Nystrom approximation it is far more difficult to do so, and it is very pleasing to see that the authors of this paper were able to do this. I believe that understanding this regularisation effect is very important for large scale algorithm design in the future.

The paper is well written, well structured, concise and to the point. The experiments are convincing.

Some very minor points: - page 2: ALS: adding a reference would be useful - page 5 before sec. 3.3: suggests --> suggest - page before Sec 4: On reading it took me some time to figure out that the theoretical setting of the parameters lambda and m is not what the algorithm will follow. If this would get across sooner (and the reasons for it) it would help the reader. - a conclusions section is missing, although I can see that it would be rather difficult to squeeze it in. - the references seem to be in a random order. Alphabetic order would be more helpful.

Summary: The paper gives optimal error bounds for 2 Nystrom-approximated schemes kernel ridge regression in the random design setting, as well as experimental validation on 3 benchmark data sets.

Submitted by Assigned_Reviewer_4

Summary: The authors study the computational/statistical tradeoff of Nystrom methods in the kernel ridge regression problem in case of random design. They derive optimal finite sample bounds for the excess risk of the estimator (Theorem 1) by choosing the (n,m,\lamda) [n - number of samples, m - number of subsamples in the Nystrom method, \lambda -regularization parameter] appropriately. The paper also gives insight how 'm' plays the role of regularization both from theoretical (computational error term) and practical point of view (numerical experiments). The proof relies on approximation theory and functional analysis; the combination of these results to obtain the bounds is nice and technical. The paper is well organized, the results are clearly presented and relevant.

Below I enlist my major comments, improvement suggestions; my minor comments are listed afterwards:

-Generally, I would encourage the authors to avoid the 'it is easy to see' type of statements: such _conjectures_ are often error-prone (some additional derivation never hurts).

main: -line 280-281: In '1)' is the second term really of the same order as the first and third one? This requirement would imply two (possibly contradictory) equations for 'm'. Please detail and clarify the statement. -line 338-342: The stated update rule via the block matrix inversion lemma is screwed up (in \alpha_t: M_t:(tl)x(tl), C_t:nx(tl), \hat{y}_n: nx1 -- they are not compatible; C_1 is undefined; in D_t, the last B^t could be B_t, ...). Please, provide a precise derivation and formula.

supp: -Lemma 3: In the proof of Lemma 3 Prop. 11 is applied, which requires H=H(K) to be a _separable_ Hilbert space. An RKHS is not necessarily separable; thus, assume/guarantee it. A relatively mild sufficient condition is 'separable topological domain with a continuous kernel'; see Lemma 4.33 in [25]. -Lemma 1: --\alpha comes from line 559, thus in line 555 \alpha should be \tilde{\alpha} and \hat{y} should be \hat{y}_n [in accord with Eq. (12)]. From the first term (=pseudo-inverse) an 1/n multiplier appears, from B_{mn} an \sqrt{n} => an \sqrt{n} term seems to be missing from the final expression. ? --line 549: This \hat{y}/\hat{y}_n definition is unclear: \hat{y}_n has already been defined (line 98)... -line 858: Based on the B1 bound, the exponent of C_m in Eq. (15) seems superfluous (i.e., it is equal to 1); the error propagates to line 928 and 963.

Minor comments/typos: ==================== main: -||operator||: define/denote that this is the operator norm (not Hilbert-Schmidt norm/...); the same holds for vector norms (it appears in the text as ||v|| and ||v||_{H} as well; I would recommend to use the 2nd form) -line 88: \lambda \ge 0 -> \lambda > 0 (otherwise for example uniqueness might not hold) -line 113: The (K_{mn})_{ji} = K(\tilde{x}_j,x_i) (or K_{mn}=K{nm}^T) definition is missing -line 267: \hat{f}_{\lambda m} -> \hat{f}_{\lambda,m} (comma is missing, see also throughout the paper) -line 280: approah -> approach -line 319: Eq.5 -> Eq.~(5); see also line 374. -line 319: the selected points -> be the selected points -line 320: dimension l -> size l -line 320: please add "t=1,...,T" -line 338: \alpha_t should be \tilde{\alpha}_t -line 344: to compute \tilde{a}_1,...,\tilde{a}_T -> to compute \tilde{\alpha}_1,...,\tilde{\alpha}_T -References: [9,14,18]: the accents are missing from Nystrom

supp: -line 515: the this -> this -line 532: ']H' -> _{H} -line 533: Z_m* (the '*' is missing on the left) -line 541: bold K -> K_n (see under Eq. (4)) -line 541: "B_{nm} = \sqrt{nm}S_n Z_m*" -> "B_{nm} = \sqrt{n}S_n Z_m*" ('m' seems to be superflous) -line 541: "G_{mm} = m Z_m Z_m*" -> "G_{mm} = Z_m Z_m*" (m is not needed) -line 542: bold \tilde{K} -> \tilde{K}_n [see Eq. (13)] -line 571: UU* should be U*U (it occurs twice) -line 597: S_n should be S_n*; the same holds in line 604, 861 -line 622: \sqrt{()^2} is superfluous -line 636-638: partition -> subset (twice); see also line 650 -line 644: C_m should be "H -> H (or H_m)" type mapping, not "H -> R^m" -line 646: C_{\lambda} has not been defined (only later) -line 663: the the -> the -line 719: Therfore -> Therefore -line 725: colum -> column -line 855: the accent is missing from 'Nystrom' -line 884: What is r? -line 895-896: The 's' character is already in use (see Assumption 4), please select an other one. -line 935: taking -> take

-line 1050: "let denote || ||_HS" -> "let || ||_HS denote" -line 1052: "Let name" -> "Let"; the same holds in line 1063 -line 1087: "Let n, n ... integers" -> "Let n ... integer" -line 1099: ". For any" -> ", then for any" -Please add fullstops (for example) to line 578, 598, 632, 985, 997, and use commas in line 604, 858, ...
Summary: The authors focus on the problem of understanding the statistical aspects (generalization properties) of Nystrom type subsampling schemes in kernel ridge regression when the design is random. The main contribution of the paper is Theorem 1: these methods can implement minimax optimal rates with 'small' memory footprint; although the paper contains a few errors, the general idea seems to be correct, the presentation is clear and the contribution is significant.

Author Feedback
Author rebuttal: First, we would like to thank the reviewers for their careful reading and thoughtful comments.

The reviewers have kindly included corrections to typos, suggested relevant references and described how some parts can be improved and presented in a more clear way. We will carefully follow their suggestions in a possible final version.

Here, according to the text limits, we discuss the major comments:

* Assumption 1
In general Assumption 1 is not superfluous, indeed \cal{E} may not have a minimum in H (as pointed out in [26] or in "Learning from Examples as an Inverse Problem", De Vito, Rosasco et al.). In [28] Ass.1 is substituted by other assumptions on H and on the noise (see next comment).
We will discuss a bit more about it in the paragraph on line 148.

* Comparison of our assumptions with the ones of [26, 28] and the role of lowerbounds
For a discussion on minimax and individual lower bounds see Sec. 2 of [26], here we note that while the upper bounds depend on the specific learning algorithm, the lower bounds depend only on the distribution of the data and are algorithm-independent. The paper [26] provides minimax and individual lower bounds for a setting subsumed under ours, indeed our Ass. 1,3,4 are equivalent to theirs (Eq. 4,5,6,7,8 plus ii, iii of Def.1), while their Eq. 9 is stronger than our Ass. 2. Therefore their lower bounds still hold in our setting and make our upper bound optimal.
The paper [28] provides optimal rates for plain KRLS in a different setting that partially intersects with the one presented in [26]. The setting in [28] does not need Ass. 1, but a stronger assumption on H (Eq. 7) and on the noise (Eq. 3) are required.
We will expand lines 235-238 to include part of the discussion above.

* Clarification of lines 280-281
In lines 280-281 we tried to explain what we did in the proof of Thm.1. As pointed out by Reviewer 3, the point 1 and 2 are switched. Term 1 of Eq. 10 is antitone w.r.t. lambda while the third is monotone, the second term is antitone in m. Thus we first optimize lambda in order to minimize Term 1 + Term 3, then we choose an m such that C(m) <= 3lambda. Hence Term 2 becomes of the same order of Term 3. In this way the l.h.s. of Eq. 10 is bounded by a constant times minimum of Term 1 + Term 3.
We will enhance lines 280-281.

* More info on the experimental setting
The selected datasets are already divided in a training part and a test part. In the experiments, the test set is fixed and corresponds to the test part, while the training part is randomly splitted each time in a training set (80%) and a validation set (20%). The points for the Nystrom method are randomly selected from the training set. In the paragraph at line 353, for each dataset, the sigma of the gaussian kernel has been chosen by a preliminary holdout on plain KRLS.
We are going to write it in lines 349-352.

* page before Sec. 4, clarification on the experiments
In Thm. 1 we proved that the optimal lambda, m depend on specific properties of the problem (s and gamma). In practice such properties of the problem are unknown, therefore, to find lambda and m, we need a model selection step, for which we provided an efficient incremental algorithm. Note that, by combining a simple adaptation of Thm 7.24 in [25] with the results of our Thm. 1, we could prove that the parameters chosen by hold out give a learning rate of the same order of the optimal one (see Thm. 8 [28] and discussion above), but it is beyond the scope of this paper and we defer it to a future work.
We will add part of this comment at the beginning of Sec. 4.

Thanks to the meticulous proof checking of the reviewers, some typos in the proofs have been discovered. We will apply all the corrections suggested. Here we discuss some of them:
- The updated rule presented in lines 338-342 contains some typos, in the final version we will correct it and provide an explicit derivation.
- We missed to make explicit the standard requirement for H to be separable. To guarantee this condition we apply the correction suggested by Reviewer 3 at the beginning of Section 2.
- We will remove the typo on \hat{y} and \hat{y}_n that make the definition unclear. In particular we are going to define \hat{y} = (y_1,...,y_n) on line 97 and \hat{y}_n = 1/sqrt(n) \hat{y} on line 548, before Lemma 1. Therefore in Eq. 4,5,12,13 \hat{y} will be substituted by \hat{y}_n, while we are going to keep unchanged the proof of Lemma 1 and the subsequent Lemmas.
- As pointed out by Reviewer 3, there is a typo on the exponent of the definition of C(m) in line 861. The correct definition is C(m) = ||(I-Pm)(C+lambda)^1/2||^2, while we keep Eq. 15 unchanged.
- We will avoid the use of 'it's easy to see', by providing additional derivations when needed.